# Light-field flow cytometry for high-resolution, volumetric and multiparametric 3D single-cell analysis

Xuanwen Hua [1,5], Keyi Han [1,5], Biagio Mandracchia [1], Afsane Radmand[2,3], Wenhao Liu[1], Hyejin Kim [1], Zhou Yuan[2,4], Samuel M. Ehrlich[2,4], Kaitao Li[1], Corey Zheng[1], Jeonghwan Son[1], Aaron D. Silva Trenkle[1], Gabriel A. Kwong[1,2], Cheng Zhu[1,2], James E. Dahlman [1,2] & Shu Jia [1,2] ✉

Imaging flow cytometry (IFC) combines flow cytometry and fluorescence microscopy to enable high-throughput, multiparametric single-cell analysis with rich spatial details. However, current IFC techniques remain limited in their ability to reveal subcellular information with a high 3D resolution, throughput, sensitivity, and instrumental simplicity. In this study, we introduce a light-field flow cytometer (LFC), an IFC system capable of high-content, single-shot, and multi-color acquisition of up to 5,750 cells per second with a near-diffraction-limited resolution of 400-600 nm in all three dimensions. The LFC system integrates optical, microfluidic, and computational strategies to facilitate the volumetric visualization of various 3D subcellular characteristics through convenient access to commonly used epi-fluorescence platforms. We demonstrate the effectiveness of LFC in assaying, analyzing, and enumerating intricate subcellular morphology, function, and heterogeneity using various phantoms and biological specimens. The advancement offered by the LFC system presents a promising methodological pathway for broad cell biological and translational discoveries, with the potential for widespread adoption in biomedical research.

Flow cytometry and fluorescence microscopy are two vital and informative driving forces for biological and medical research. Flow cytometry allows for the rapid analysis of diverse cellular populations, while fluorescence microscopy provides a high-resolution image of individual cells. The emergence of imaging flow cytometry (IFC) combines these strengths, enabling high-throughput, multiparametric single-cell analysis with rich spatial details, high sensitivity, and molecular specificity[1–4]. The ability to acquire cytometric images allows for the direct visualization of cell properties, such as size, shape, biomarker intensity, physiological state, and other morphological and biochemical characteristics[5]. IFC technologies have been applied across various basic and translational fields, including cell biology[6], immunology[7,8], microbiology[9], hematology[10], and cancer research[11,12].

Significant advancements have been made in cytometric imaging capabilities, such as speed, sensitivity, and resolution, through the integration of various fluorescence microscopy strategies[13–19]. However, current IFC systems, in comparison with other single-cell imaging platforms, remain disadvantageous in the data acquisition at higher resolution and dimensions[3]. While some IFC approaches have achieved sub-micrometer resolution with high throughput[14,15], they primarily

[1]Wallace H. Coulter Department of Biomedical Engineering, Georgia Institute of Technology and Emory University, Atlanta, GA, USA. [2]Parker H. Petit Institute for Bioengineering and Biosciences, Georgia Institute of Technology, Atlanta, GA, USA. [3]Department of Chemical Engineering, Georgia Institute of Technology, Atlanta, GA, USA. [4]Georgia W. Woodruff School of Mechanical Engineering, Georgia Institute of Technology, Atlanta, GA, USA. [5]These authors contributed equally: Xuanwen Hua, Keyi Han. ✉e-mail: shu.jia@gatech.edu

generate 2D cell images, consequently losing crucial 3D spatial information. Alternatively, the 3D subcellular image acquisition has been proposed based on relevant microscopy techniques, such as light-sheet microscopy[20], confocal microscopy[21], beam engineering[22,23], and tomography[24,25]. Nonetheless, these methods may necessitate compromises between 3D resolution, volumetric coverage, and throughput due to sequential acquisition, which may also lead to increased instrumental complexity and limited accessibility on commonly used platforms such as epi-fluorescence microscopes. As a result, IFC-based platforms for single-cell investigations have yet to achieve the optimum balance in uncovering 3D subcellular details with high resolution, throughput, sensitivity, and uncomplicated instrumentation.

The advent of light-field microscopy (LFM), on the other hand, presents a particularly appealing solution for capturing fast-moving single-cell specimens. In essence, LFM can concurrently record the spatio-angular information of light, enabling computational reconstruction of the volume of a biological sample using just a single camera frame[26–36]. Recent advancements in Fourier LFM (also known as extended LFM) have further improved the image quality and computational efficiency[37–40], facilitating 3D subcellular, millisecond spatiotemporal studies across various biological systems, such as the functional brain[38,41], organoids[42], and single-cell specimens[43,44]. In comparison to other 3D techniques, the light-field approach promises single-shot, scanning-free 3D acquisition and instrumentally simple operation on epi-fluorescence platforms, both of which are highly desirable features for cytometric imaging.

In this study, we introduce a light-field flow cytometer (LFC), an IFC system designed for 3D volumetric, high-throughput, and multi-parametric analysis of single-cell populations. The LFC system incorporates a high-resolution light-field optofluidic platform, hydrodynamic focusing, and stroboscopic illumination, offering a near-diffraction-limited and multi-color resolution of various 3D subcellular morphologies across all three dimensions at high speeds. We demonstrate the system by examining and quantifying a range of phantoms and biological morphologies, functions, and heterogeneities, including peroxisomes and mitochondria in cultured cells, morphological characterizations of isolated cells from mice and humans, apoptotic alterations in staurosporine-treated Jurkat cells, and the expression of tdTomato following Cre mRNA delivery in mice. We expect LFC, as an accessible and compatible cytometric imaging technique, to significantly advance cell biology and translational research.

## Results

### Light-field flow cytometry

As depicted in Fig. 1a, the LFC system was constructed based on a high-resolution epi-fluorescence platform, which incorporates a 100×, 1.45 numerical aperture (NA) objective lens and an array of optical configurations (see Methods and Supplementary Fig. 1 for a detailed schematic). In particular, the epi-fluorescence image at the native image plane was Fourier transformed and partitioned by a customized hexagonal microlens array (MLA) (Supplementary Note 1), forming elemental light-field images on the back focal plane of the MLA, which were captured by an sCMOS camera[43]. To ensure consistent cell occupancy within the light-field acquisition volume, hydrodynamic focusing was implemented into the microfluidic system by sheathing the sample with faster flows[45] (Fig. 1b, Supplementary Note 2). Furthermore, stroboscopic illumination with coaxial laser lines (488 nm, 561 nm, and 647 nm) was generated by function-generating devices[13], allowing for multiple single- or multi-color exposures within a single camera frame while eliminating motion blur at high flow speeds (Fig. 1c, Supplementary Note 3).

The captured elemental light-field images can be considered a convolution between the light-field point-spread function (PSF) and the object volume, which allows for the 3D retrieval of the object through an inverse computational process[40] (Fig. 1d–f). Specifically, the elemental images were first processed using a lab-written denoising algorithm ACsN[46] to enhance sensitivity under low signal-to-noise (SNR) conditions resulting from short exposure times. Then, the images underwent wave-optics-based 3D deconvolution with a hybrid point-spread function, facilitating accurate volumetric reconstruction calibrated for system deviations while minimizing computational artifacts throughout the entire imaging depth[42,43] (Fig. 1f, Supplementary Fig. 2, and Supplementary Notes 4 and 5). By integrating optical, microfluidic, and computational strategies, unlike other high-resolution 3D optofluidic imaging strategies[47–49] that suffer low throughput (typically 10-20 cells/sec), the LFC system enables blur-free and volumetric visualization of various 3D subcellular morphologies at high speeds, reaching up to 5,750 cells/sec while maintaining a high SNR (Supplementary Note 6).

### Characterization of LFC with phantom samples

To characterize the LFC system, we initially imaged phantom samples within the flow and assessed the 3D reconstructed multi-color images (Fig. 2). Specifically, we used a mixture of Tetra-Speck fluorescent microspheres with diameters of 200 nm, 1 μm, 2 μm, and 4 μm. These microspheres were injected at a flow rate of 0.4–0.6 μL/min (approximately 4.50 mm/sec, 5,000–10,000 objects/sec), hydrodynamically focused, stroboscopically excited by three laser lines with 100-μs illumination durations and recorded at 200 frames per second (fps). The reconstructed microspheres displayed a range of volumes within the flow (Fig. 2a). In particular, the 3D images of sub-diffraction-limited 200-nm microspheres exhibited the full width at half maximum (FWHM) values at 337 nm, 291 nm, and 542 nm in the $X$, $Y$, and $Z$ dimensions, respectively (Fig. 2b). Moreover, the 3D measurements of different phantoms aligned well with the physical profiles of the samples convolved with the expected near-diffraction-limited 3D resolution in the lateral and axial dimensions, respectively (Fig. 2b–e, Supplementary Note 7). Additionally, a > 5× extended depth of focus (~6 μm) was observed for high-resolution light-field acquisition compared to conventional epi-fluorescence microscopy (Supplementary Note 7). The measurements of the reconstructed objects revealed four distinct populations, in which both the microsphere diameters derived from the 3D volumes and the corresponding intensity matched the expected values. The results demonstrated that the LFC system can reliably identify each sub-population within the phantom mixture (Fig. 2f–k, Supplementary Figs. 3, 4, and Supplementary Movie 1).

### Multi-color imaging of peroxisomes and mitochondria in flowing HeLa cells

To demonstrate 3D subcellular imaging, we first analyzed flowing HeLa cells labeled with peroxisome-GFP using 488-nm laser excitation (Fig. 3a–d). The cells were introduced at a relatively slow flow rate of approximately 0.03 μL/min (~0.11 mm/sec), and the GFP signals emitted by peroxisomes were captured without motion blur at 200 fps under continuous illumination. The reconstructed light-field image, based on a single camera frame, revealed the intricate 3D structures of peroxisomes prominently distributed across a cellular thickness over 3 μm (Fig. 3a, b, Supplementary Fig. 5a). The vesicles, separated by as close as 400–600 nm, could be well-resolved in all three dimensions (Fig. 3c, d). Furthermore, we conducted two-color imaging of mitochondria and peroxisomes in flowing HeLa cells labeled with MitoTracker and peroxisome-GFP, respectively (Supplementary Fig. 6, Supplementary Movies 2 and 3). The LFC system captured the optical signals of both organelles at a flow rate of 0.4 - 0.6 μL/min (~3.4 mm/sec) using 488- and 647-nm lasers alternatively with 100-μs stroboscopic illumination duration at 200 fps (Fig. 3e, f, Supplementary Fig. 5b, c). The flowing cells displayed a

native, sphere-like morphology, and the reconstructed two-color images depicted the intricate 3D spatial relationship between peroxisomes and mitochondria across a significant thickness (~6 μm) of the cells (Fig. 3g, h). Remarkably, the high resolution and volumetric capabilities enabled visualization of structural variations as close as 400–600 nm for both organelles in all three dimensions (Fig. 3i, j),

consistent with the resolution measurements obtained using the phantom samples. These results, enhanced by effective denoising, displayed reliable reconstruction under varying SNR conditions (Supplementary Fig. 7) and high accuracy compared with other modalities, such as epi-fluorescence and 3D structured-illumination microscopy (SIM) (Supplementary Note 8).

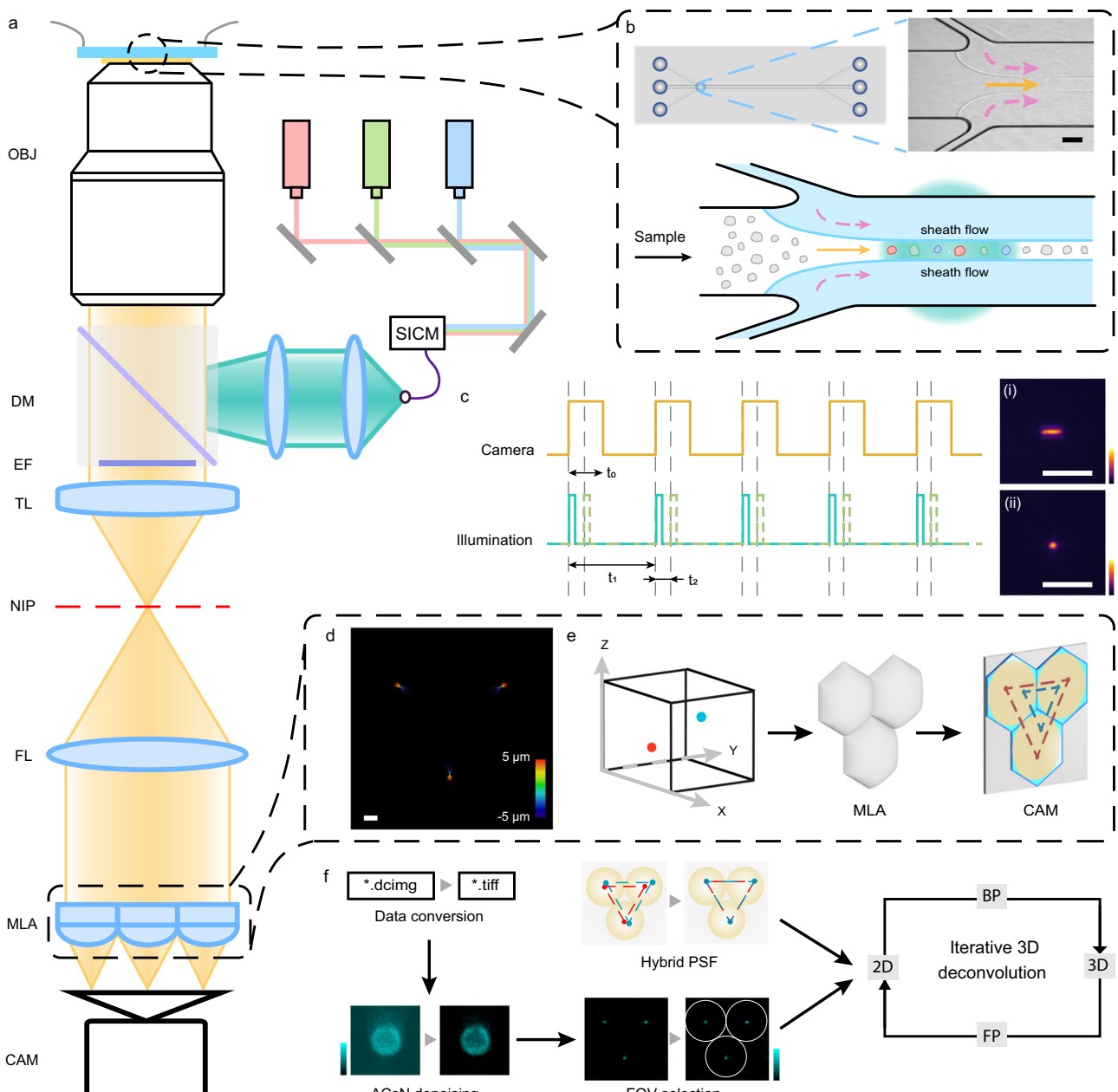

**Fig. 1 | Light-field flow cytometer (LFC). a** Schematic of the LFC system. Laser lines, modulated by a stroboscopic-illumination controlling module (SICM) and reflected by a dichroic mirror (DM), excite samples in microfluidic flow. An oil-immersed objective lens (OBJ), emission filter (EF), and tube lens (TL) create wide-field images at the native image plane (NIP). A Fourier lens (FL) optically transforms the NIP onto its back focal plane, where a microlens array (MLA) partitions the light field to generate three elemental images onto an sCMOS camera sensor (CAM) located at the back focal plane of the MLA. **b** Microfluidic setup. The microfluidic chip (top) contains a main sample channel (solid arrows; width = 500 μm, depth = 30 μm) with two side channels injected with red HBSS (dashed arrows). The pressure difference is adjusted to create the proper hydrodynamic focusing for the sample solution (top inset and bottom) of 70–80 μm in width, agreeing with

the FOV of the imaging system. **c** Stroboscopic illumination of laser lines is synchronized and controlled within each digital camera exposure to minimize motion blur (insets i and ii). Multiple illumination cycles can be generated within each global camera exposure $t_0$ at an interval of $t_2$. The period of the multi-illumination cycle sets $t_1$ corresponds to the camera frame rate. **d** Axial stack projection (step size = 100 nm) of the hybrid point-spread function (hPSF) through the customized MLA within an axial range of 10 μm, as color-coded in the color scale bar. **e** Light-field image formation for emitters at different 3D positions, capturing both the spatial and angular information in an uncompromised manner. **f** Image processing pipeline, containing image conversion, ACsN denoising, elemental image selection, and deconvolution-based image reconstruction using the hybrid PSF. Scale bars: 100 μm (**b**), 5 μm (**c**). 10 μm (**d**).

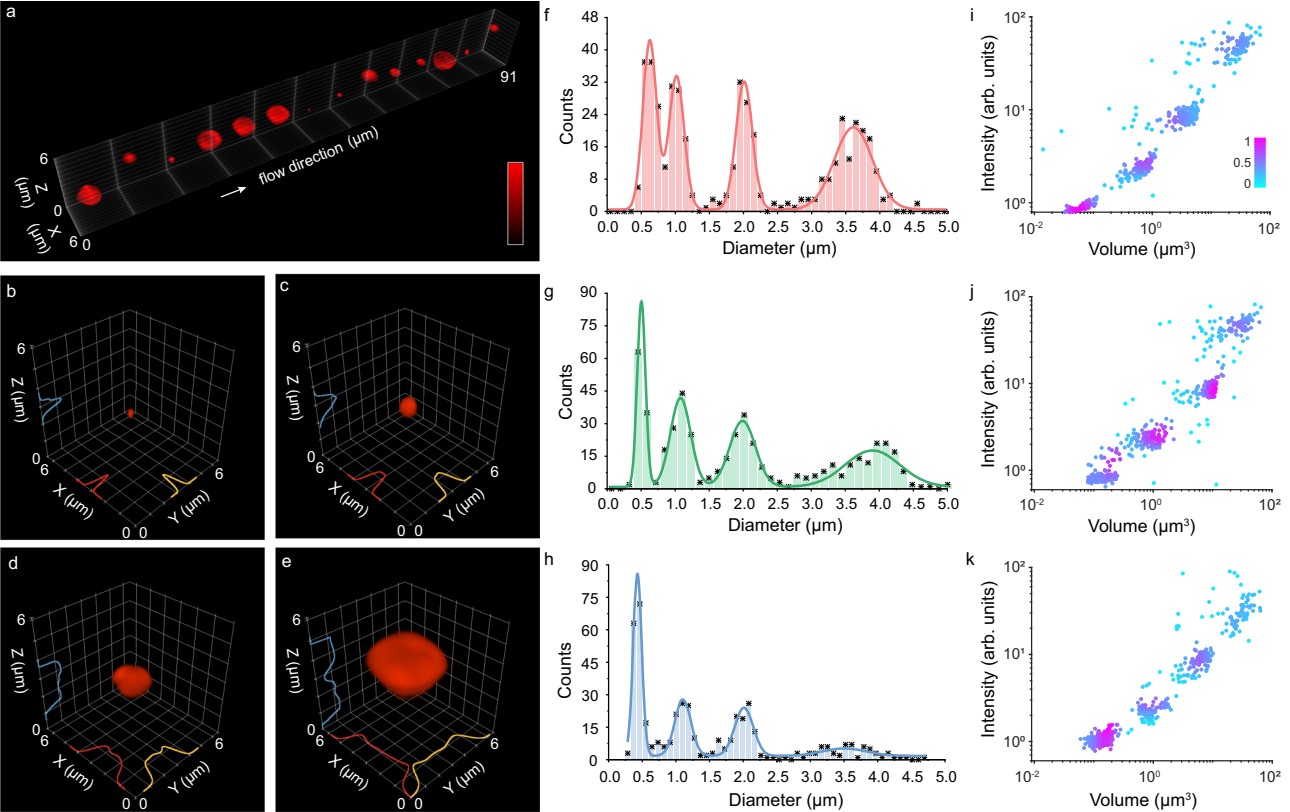

**Fig. 2 | Characterization of LFC using fluorescent microspheres. a** A mixture of deep-red TetraSpeck fluorescent microspheres with diameters of 200 nm, 1 μm, 2 μm, and 4 μm. 3D reconstructed images of the microspheres with intensity profiles along three axes, exhibiting FWHM values of 337 nm, 291 nm, 542 nm for 200-nm (**b**), 826 nm, 743 nm, 1014 nm for 1-μm (**c**), 2002 nm, 2069 nm, 2587 nm for 2-μm (**d**), 3891 nm, 3989 nm, 4696 nm for 4-μm (**e**) microspheres in X, Y, Z, respectively. The high SNR of microspheres enables a spatial resolution measurement between 300–600 nm. Histogram counts of the microsphere diameters

rendered based on the measured 3D volumes for multi-color excitations at 647 nm (**f**) (n = 461), 561 nm (**g**) (n = 437), 488 nm (**h**) (n = 442), showing consistent distributions of spectral channels with the known microsphere diameters of 200 nm, 1 μm, 2 μm, and 4 μm. **i–k** Corresponding scatter plots of the fluorescence intensity as a function of the microsphere volumes in (**f–h**), respectively, displaying distinct four populations of microspheres. The color gradient in the scatter plots (**i–k**) serves to visualize the density distribution of the beads based on their respective volumes and intensities. Source data are provided as a Source Data file.

## Analyzing morphological features of isolated cells from mouse and human

We subsequently demonstrated 3D cytometric imaging of heterogeneous cell populations using the LFC system (Fig. 4, Supplementary Figs. 8 and 9). Specifically, we analyzed membrane-labeled blood cells (Fig. 4a–c, Supplementary Movies 4 and 5) and spleen cells (Fig. 4d–f, Supplementary Movies 6 and 7) extracted from adult mice, which were introduced at a rate of approximately 600 cells/sec. The imagery obtained displayed high specificity and sensitivity for differentiating various 3D morphological features (Supplementary Note 9) and quantifying their staining intensity on a cell-by-cell basis with a high throughput ~2,300 cells/sec (Fig. 4a–f, Supplementary Fig. 9). Next, we labeled and imaged the membrane and nucleus of mouse naïve T cells (Fig. 4g–I, Supplementary Fig. 5d, e) and human activated T cells (Fig. 4j–n) and imaged these samples at a rate of ~300 cells/sec. We applied alternating 488-nm and 647-nm illumination, each with a 100-μs stroboscopic illumination time, and captured images at 200 fps. We were able to quantify the hollow structures of the cell membrane in all three dimensions, which were shown to enclose the nucleus of each cell (Fig. 4h, i). The membrane stain exhibited a consistent thickness of 500–600 nm across three dimensions, in agreement with the measured 3D resolution of the LFC system (Fig. 2b–e). Moreover, the reconstructed focal stacks of the human-activated T cells displayed two distinct sizes of the cell membrane (7.99 μm) and nucleus (6.57 μm) (Fig. 4k–m). Notably, the volumetric capability of LFC considers diverse 3D cell morphologies, thereby facilitating accurate 3D

cellular quantification, unlike the estimations derived from 2D wide-field images that assume a spherical cell shape[50]. As a result, we were able to identify the nuclear-to-cytoplasmic (N:C) ratio of immune cells by directly measuring the ratio of nuclear volume to total cell volume, which exhibited a mean N:C ratio of 0.55 (Fig. 4n), consistent with the previously reported results[13,15,51].

## Imaging morphological changes in staurosporine-treated Jurkat cells

Programmed cell death is a crucial stage for proper tissue and organ functioning, and its malfunction is often associated with various diseases[52,53]. Apoptosis, one of the primary pathways of programmed cell death, involves numerous morphological and functional changes inside cells, such as chromatin condensation, nuclear fragmentation, loss of cell contact, and organelle swelling[54]. Staurosporine (STS), a protein kinase inhibitor isolated from *Streptomyces*, has been widely used to induce apoptosis in various types of cells[55]. Since apoptosis occurs in a 3D manner in the cellular space, a flow cytometer with 3D imaging capability is essential for observing apoptotic status with higher sensitivity and accuracy. In this study, we demonstrated LFC to investigate 3D subcellular morphological alterations of human T lymphocyte (Jurkat) cells arising from STS-induced cell apoptosis. Experimentally, we conducted cytometric imaging of Jurkat cells after treatment with 1-μM STS for 30, 60, 120, and 300 min. In comparison with wide-field images (Supplementary Fig. 10 and Supplementary Movie 8), multi-color LFC captured multiple organelles of flowing cells,

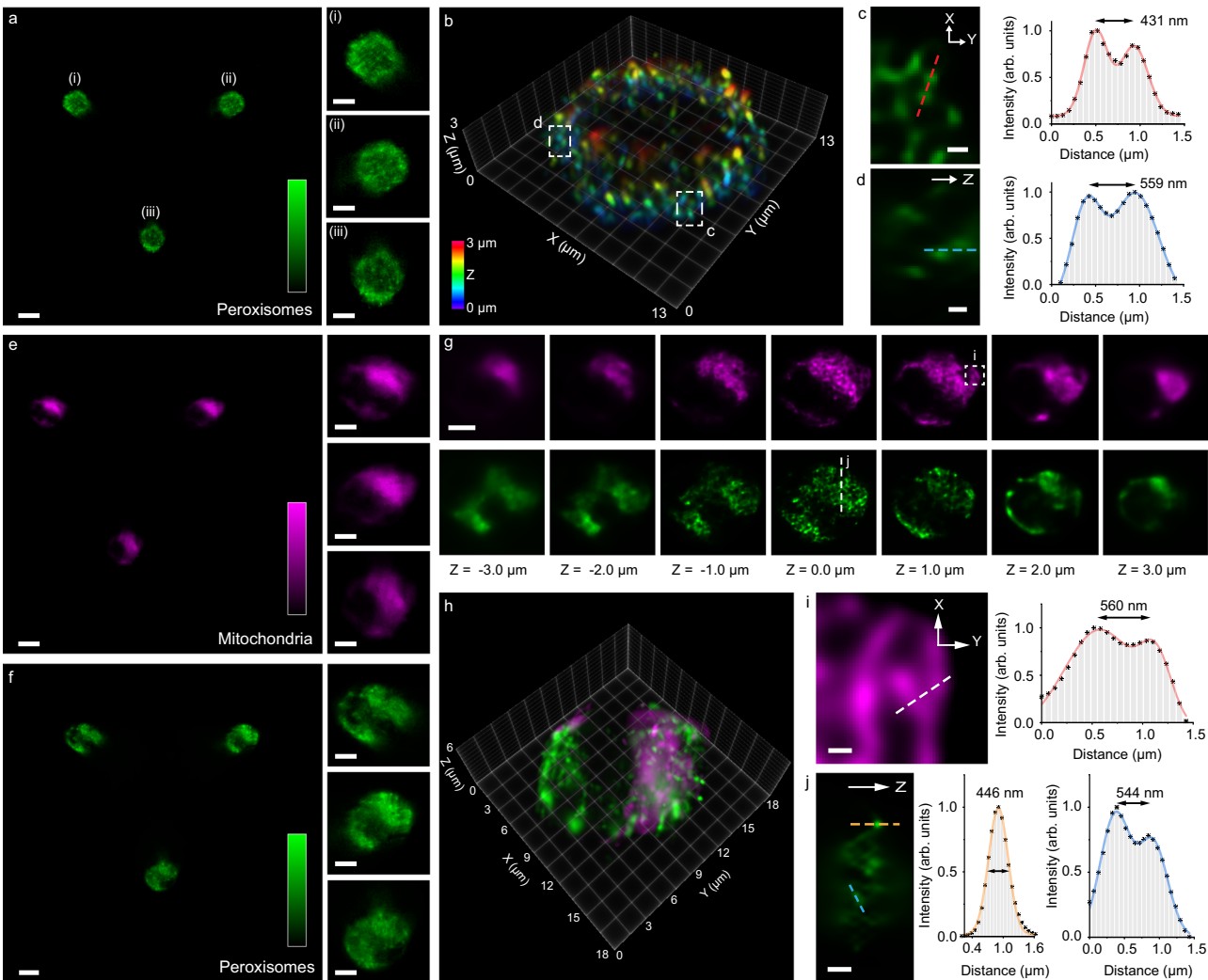

**Fig. 3 | Imaging peroxisomes and mitochondria in flowing HeLa cells with LFC.**
Denoised LFC image (**a**) and 3D reconstructed volume (**b**) of peroxisomes in HeLa cells. The LFC image in **a** is representative of >5 cell images acquired under identical experimental conditions. Insets (i-iii) in (**a**) show zoomed-in elemental images. Zoomed-in images in X-Y (**c**) and Z (**d**) of the corresponding boxed regions in (**b**), showing nearby peroxisomes resolved as close as 400–600 nm in all three dimensions. Denoised two-color LFC images of mitochondria (**e**, magenta) and peroxisomes (**f**, green), their corresponding reconstructed axial stacks (**g**), and

merged 3D image (**h**) of HeLa cells. Insets in (**e**) and (**f**) show the corresponding zoomed-in elemental images. Zoomed-in images in X-Y (**i**) and Z (**j**) of the corresponding regions indicated in (**g**), showing resolved subcellular structures of mitochondria and peroxisomes as close as 400–600 nm in all three dimensions. The LFC images in (**e**) and (**f**) are representatives of >100 cell images acquired under identical experimental conditions. Scale bars: 10 μm (**a**, **e**, **f**), 5 μm (**a** insets, **e** insets, **f** insets, **g**), 500 nm (**c**, **d**, **i**, **j**). Source data are provided as a Source Data file.

such as the nucleus and mitochondria, with high resolution and clarity, enabling the visualization of the 3D morphology of their subcellular organizations (Fig. 5a–c and Supplementary Movies 9–13). These apoptotic morphological changes have also been validated using epifluorescence and 3D SIM (Supplementary Note 10). Notably, the previously spherical and intact nuclei underwent significant morphological changes, exhibiting fragmented and condensed nuclear dispersion throughout the cell, a characteristic feature of cells undergoing apoptosis[56] (Fig. 5d–l). With the treatment period increased, the Jurkat cell nuclear morphology displayed reduced volumes and fragmented micronuclei. Consequently, over 53% of the cells showed apoptotic nuclei after 5 h of treatment (Fig. 5m–p and Supplementary Fig. 11). Meanwhile, during the dispersion of fragmented nuclei within the cellular volume, organelles such as mitochondria experienced an increased degree of enclosure amidst the interstitial spaces of the micronuclei (Fig. 5q and Supplementary Fig. 12). These results underscore the utility of LFC in elucidating the delicate subcellular morphological alterations associated with various cell functions and

dysfunctions within 3D volumetric, multiparametric, and population-based context.

## Image-based analysis of tdTomato⁺ expression after Cre mRNA delivery

Lipid nanoparticles (LNPs) carrying mRNA have been used in two COVID vaccines and earlier-stage clinical trials that have generated promising results[57]. However, visualizing and quantifying functional mRNA delivery (i.e., the subsequent protein expression) has been challenging. To evaluate whether 3D IFC could achieve this goal, we formulated a liver-targeting LNP[58] so it carried mRNA encoding Cre recombinase using microfluidics[59]. We then intravenously administered the LNPs to Ai14 mice at the clinically relevant[60] dose of 0.25 mg/kg. In these mice, functional mRNA delivery leads to Cre protein, which then translocates into the nucleus, leading to the expression of tdTomato (Fig. 6a). Three days after LNP administration, we used LFC to assess the expression of tdTomato in the liver, spleen, and lung after Cre mRNA delivery in Ai14 mice. As observed, two-color LFC offered

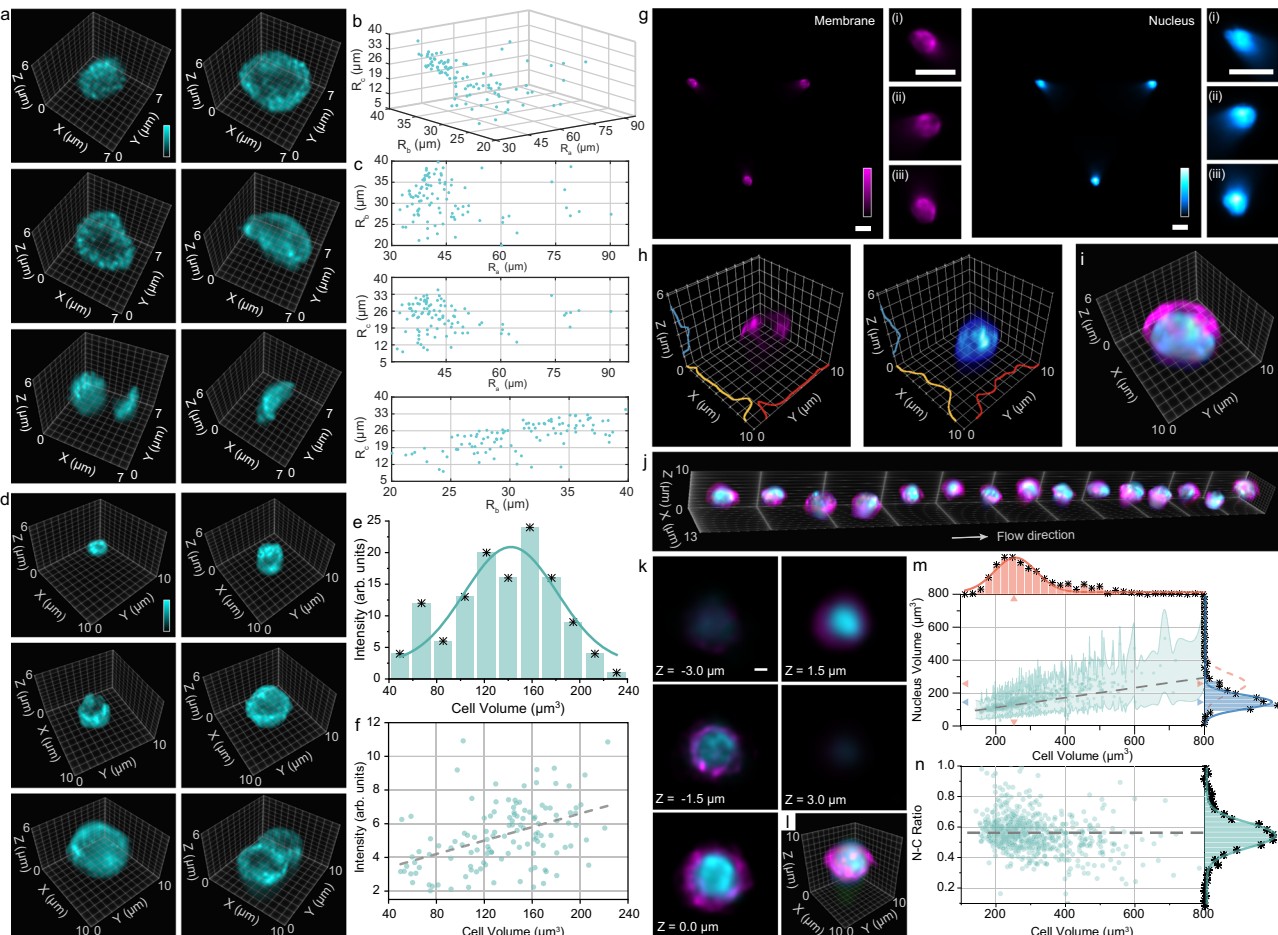

**Fig. 4 | Comparative analysis of cell morphologies in isolated mouse and human cells. a** 3D reconstructed images displaying a variety of shapes in membrane-labeled mouse blood cells. Ellipsoid-fitted radii of cells $R_a$, $R_b$, $R_c$ ($R_a > R_b > R_c$), viewed in 3D (**b**) and 2D projections (**c**), categorizing cells based on their morphologies ($n = 188$). **d** 3D reconstructed images displaying a variety of sizes in membrane-labeled mouse spleen cells. **e** Histogram of cell volumes ($n = 113$), showing an average cell volume of approximately 140 μm³. **f** Scatter plot correlating cell fluorescence intensity with cell volumes ($n = 125$). **g** Denoised light-field images of the membrane (left) and nucleus (right) of mouse naïve T cells. Corresponding insets show the zoomed-in elemental images. These LFC images are representatives of >5 cell images acquired under identical experimental conditions.

3D reconstructed images of the membrane (**h**, left) and nucleus (**h**, right) with intensity profiles along three axes and corresponding two-color overlay (**i**). **j** 3D reconstructed volumes of the membrane (magenta)- and nucleus (blue)-labeled human activated T cells in flow. Axial stacks (**k**) and two-color overlay (**l**) of a human-activated T cell in (**j**) across a depth range of 6 μm. **m** Intensity-to-volume plots for human activated T cells ($n = 679$), showing mean diameters of 7.99 μm (cell, red) and 6.57 μm (nucleus, blue) through Gaussian fitting and their linear relationship (gray dashed line with green-shaded errors). **n** The N:C ratio of human-activated T cells ($n = 679$) as a function of cell volume, indicating a mean ratio of 0.55. Scale bars: 10 μm (**g**), 1 μm (**k**). Source data are provided as a Source Data file.

high sensitivity for the 3D visualization of individual cells and their gene expression in different organs (Fig. 6b–g, Supplementary Fig. 13, and Supplementary Movie 14). Examining the percentage of cells expressing tdTomato (tdTomato⁺), we demonstrated that the liver cells were more efficiently targeted by LNPs with functional Cre mRNA delivery, with a tdTomato⁺ cell percentage of 79.41%, compared to 13.45% and 11.39% in the spleen and lung, respectively (Fig. 6h). Notably, these 3D image-based analyses showed consistent results (i.e., approximately 80% in the liver and less prominently, <20% in other organs) as reported using fluorescence-activated cell sorting (FACS)[58].

## Discussion

In conclusion, the LFC system significantly enhances cell analysis by enabling high-sensitivity, 3D volumetric, and multiparametric data acquisition, allowing for the comprehensive examination of sub-cellular morphology, behavior, and interactions within their native 3D contexts. This system features low instrumental complexity, making it compatible with commonly used epi-fluorescence microscopes and microfluidic devices. Notably, the Fourier light-field approach offers

flexible scalability to accommodate various acquisition requirements while retaining its 3D and single-shot capabilities. In addition, it permits the use of lower magnification objective lenses—commonly found in conventional IFC instruments—to address various sample sizes or fluidic dimensions and achieve an enhanced throughput[14]. Specifically, LFC combines the 100× objective lens with individual microlenses, formulating an effective magnification of 42.5× that enhances the throughput over a conventional 100× system and restores the near-diffraction-limited resolution through computational synthesis (Supplementary Note 6, 7 and Supplementary Movie 15). This combinatorial strategy alleviates the resolution-throughput tradeoff for IFC while retaining the unique snapshot 3D ability of light-field imaging (Supplementary Note 7). The functionality of the LFC system, such as the depth of focus and 3D resolution, can be further extended with various optical and computational frameworks[35,61–65] (Supplementary Notes 5 and 7). In particular, deep learning has evolved as a powerful approach to IFC systems[18,66,67], transforming a wide range of areas such as image processing, statistical analysis, and image-guided automation. In this context, deep neural networks present a viable alternative

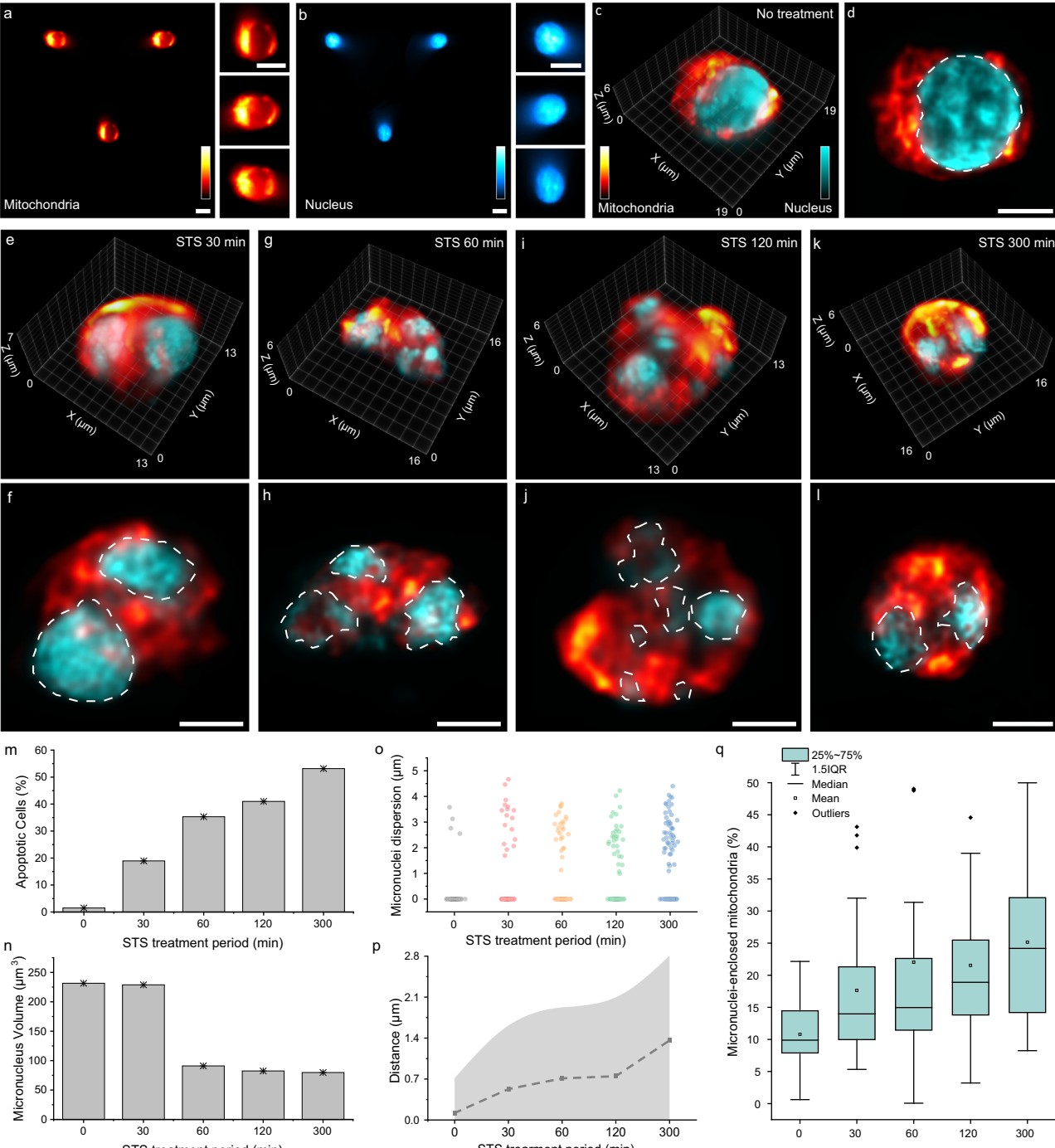

**Fig. 5 | Morphological changes in STS-treated Jurkat cells visualized through LFC.** Denoised light-field images of mitochondria (**a**) and nucleus (**b**) in a live Jurkat cell without STS treatment. Corresponding insets show the zoomed-in elemental images. These LFC images are representatives of >200 cell images acquired under identical experimental conditions. 3D reconstructed image (**c**) and one focal stack image (**d**) of the cell in (**a**) and (**b**). The dashed line indicates the nucleus segmentation from surrounding mitochondria. 3D visualization of Jurkat cells treated with STS for 30 (**e**), 60 (**g**), 120 (**i**), 300 (**k**) min and their corresponding focal stack images (**f**, **h**, **j**, **l**), respectively, exhibiting fragmented and condensed nuclear dispersion throughout the cells. **m** Percentage of the cells showing apoptotic cell morphology for each STS treatment period. **n** Average volumes of the micronuclei in cells for each STS treatment period. **o** Scatter plots displaying the average distance between individual micronuclei and their centroid for each cell for every STS treatment period. Intact nuclei with a distance of 0 μm account for 96%, 83%, 74%, 69%, and 49% of the total number of cells treated for 0 (i.e., no treatment), 30, 60, 120, and 300 min, respectively. **p** Mean (dashed) and standard deviation (shaded) of the distances in (**o**) for each STS treatment period, showing increased dispersion of fragmented nuclei. The sample size in (**m**–**p**) is 100 for each group. **q** Box plots illustrating the distribution of the volume of mitochondria enclosed within micronuclei relative to the total volume of micronuclei and mitochondria for individual cells across various STS treatment durations (*n* = 30 cells/ group). The boxes represent data from the first quartile to the third quartile. The whiskers represent data ranging within 1.5 interquartile range (IQR) values. The lines and squares within the boxes represent the medians and means for each group, respectively. The diamond data points represent outliers of the data. Scale bars: 10 μm (**a**, **b**), 5 μm (**d**, **f**, **h**, **j**, **l**). Source data are provided as a Source Data file.

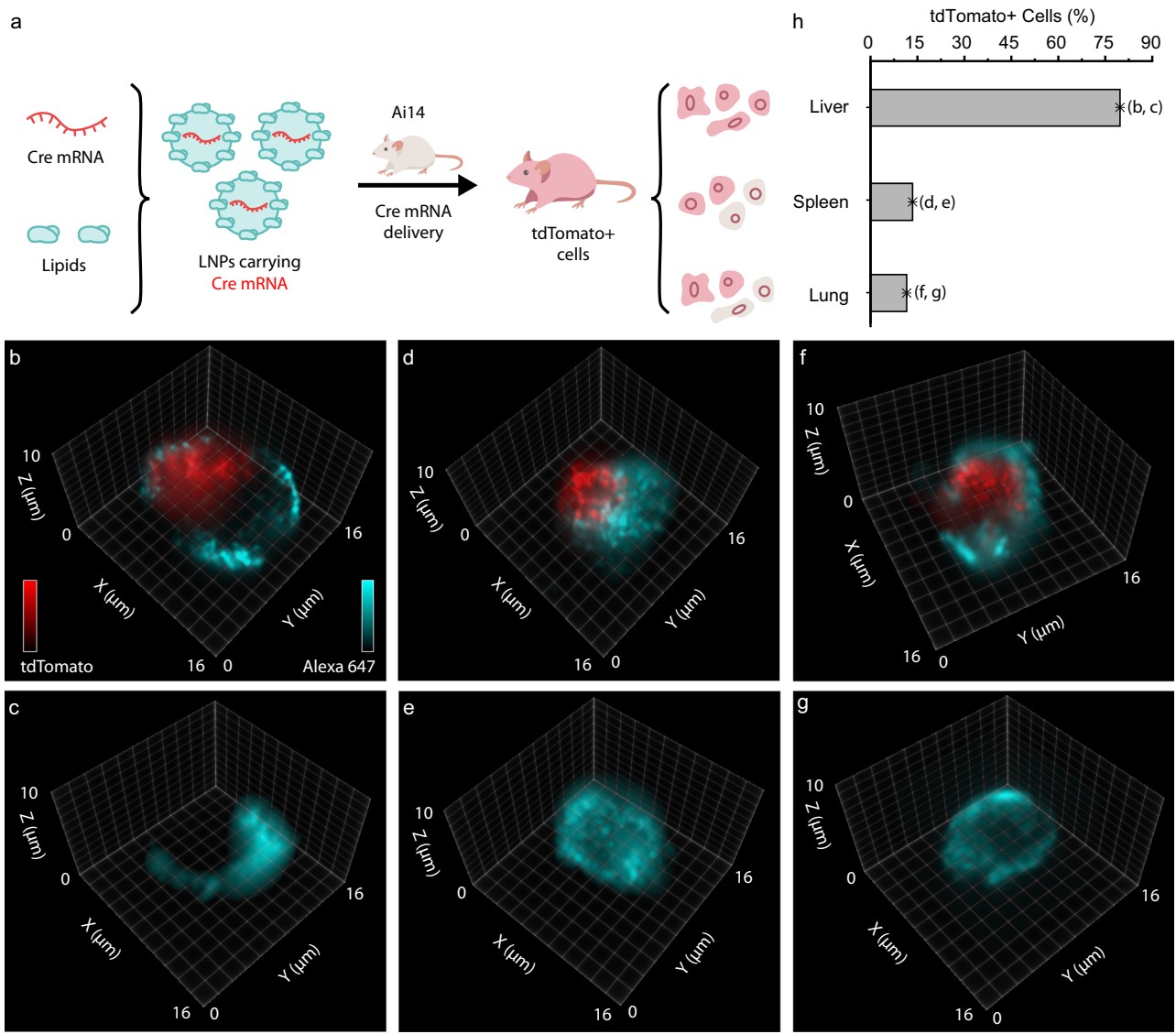

**Fig. 6 | Detection of Cre mRNA expression with lipid nanoparticle (LNP)-delivery in isolated mouse cells. a** Schematic representation of LNP formulation and delivery to cells in various organs of Ai14 mice. 3D visualization of a liver endothelial cell with (**b**) and without (**c**) the expression of tdTomato. 3D visualization of a spleen immune cell with (**d**) and without (**e**) the expression of tdTomato. 3D visualization of a lung endothelial cell with (**f**) and without (**g**) the expression of tdTomato. **h** Bar plot illustrating the percentage of tdTomato+ expression of 79.41%, 13.45%, and 11.39%, respectively, for liver ($n = 102$), spleen ($n = 316$), and lung ($n = 119$) cells. Source data are provided as a Source Data file.

to traditional deconvolution algorithms, accelerating the generation of high-quality 3D reconstructions by a factor of at least two orders of magnitude (Supplementary Note 5). Such computational efficiency holds critical implications for leveraging LFC in the cytometric analysis of large cellular populations. The approach presents great potential for broad applicability in both fundamental and translational research, with full integration possibilities that include single-cell genomics[68], microscopy-based screening and diagnosis[69,70], and image-enabled sorting[17,19]. We foresee the LFC system as a promising paradigm for a diverse array of cytometric imaging applications in fields spanning biology, pharmacology, and medical diagnostics.

## Methods
### Ethical statement
All animal experiments were performed in accordance with the Institutional Animal Care and Use Committee at Georgia Institute of Technology. Ai14, OT-I, and C57BL/6 mice were bred at the Georgia Institute of Technology Animal Facility. C57BL/6J (B6/000664) mice were purchased from Jackson Laboratories.

Human immune cells were acquired from human donors with full consent. The protocol was approved by Georgia Institute of Technology and Emory University Institutional Review Boards.

### Light-field imaging system
The high-resolution Fourier light-field microscopy system (Fig. 1a and Supplementary Fig. 1) was developed using an epi-fluorescence microscope (Eclipse Ti2-U, Nikon Instruments)[43]. The employed objective lens was an oil-immersion lens featuring 100× magnification and a numerical aperture (NA) of 1.45 (CFI Plan Apochromat Lambda 100× Oil, Nikon Instruments). A piezo nano-positioner (Nano-F100S, Mad City Labs) was utilized for precise positioning. Samples were excited using multicolor laser lines (488 nm, 561 nm, 647 nm, MPB Communications), with the fluorescence collected through a quad-band dichroic mirror (ZT405/488/561/647, Chroma) and a corresponding emission filter (ZET405/488/561/647 m, Chroma). The sample stage incorporated a micro-positioning system (MS2000, Applied Scientific Instrumentation) for accurate placement. The native image plane of the objective lens was Fourier-transformed using a

Fourier lens ($f_{FL}$ = 275 mm, Edmund Optics). A customized microlens array ($f_{ML}$ = 117 mm, RPC Photonics) was placed on the back focal plane of the Fourier lens (Supplementary Note 1). The elemental images formed by each microlens were captured using an sCMOS camera (ORCA-Flash 4.0 V3, Hamamatsu Photonics, pixel size $P_{cam}$ = 6.5 μm).

## Flow cytometer and microfluidic preparation

The microfluidic setup was constructed with a 3-channel microfluidic flow controller (OB1 MK3+, Elveflow), a microfluidic flow sensor (MFS3, Elveflow), microfluidic chips (10001824, ChipShop), microfluidic reservoirs (LVF-KPT-M-2, Darwin Microfluidics), a syringe (BD-PLSTPK-LL-01, Darwin Microfluidics), and a waste tank (Supplementary Fig. 1). Prior to the experiments, the reservoirs were filled with deionized water. During the experiments, we first opened the pump side valve (Valve 1) while blocking the syringe side valve (Valve 2) to flush the chip with deionized water from all three channels, effectively cleaning the channels before the measurements. Following the pre-experimental cleaning, we halted the pump and replaced the solution in the two small reservoirs connected to the side channels of the chip with Hank's balanced salt solution (HBSS). We then reactivated the pump to establish a stable, focused flow (Supplementary Note 2). Upon achieving the flow without bubbles in the channels, we closed Valve 1 and opened Valve 2. The samples were injected into the tubes and the chip by a syringe. Once the samples filled the tubes (1-2 mL, determined empirically), we closed Valve 2 and reopened Valve 1, allowing the samples to be automatically and controllably introduced into the chip.

## Cytometric image acquisition

After loading the samples into the microfluidic system, we initially employed the epi-fluorescence port and a 10× objective lens (CFI Plan Fluor 10×, Nikon Instruments) to monitor the entire flow due to its large field of view (FOV). At this stage, we set a high pressure to achieve a rapid flow speed, ensuring that the injected cells swiftly entered the chip channel. Upon observing the sample fluorescence, we switched to a 100× objective lens and adjusted the microscope stage to bring the sample flow into the FOV. Subsequently, we transitioned to the light-field port and commenced acquisition. The sample fluorescence was excited using stroboscopic illumination to minimize the motion blur (Supplementary Notes 3 and 6). By employing the high-speed streaming mode in the sCMOS camera, we cropped the image size to 1024 × 1024 pixels or 1024 × 900 pixels, which covered all three elemental images, depending on the synchronization needs. We then set the camera exposure time to 5 ms and initiated acquisition at a frame rate of 200 Hz for each cycle of 60,000 frames using HCImage Live 4.5.0.0 (16-bit depth, also see Supplementary Table 4). We repeated the aforementioned sample-loading step every 3-4 acquisitions to replenish the microfluidic chip with additional samples.

## Image processing

The acquired images were first converted by lab-written Python and MATLAB codes to multipage TIFF images (Supplementary Fig. 2). Then, the images were screened to exclude non-specific fluorescence from blank frames and sample debris. For multi-color imaging, the two adjacent frames were selected, and each fluorescence representing certain subcellular signals was identified and sorted into separate folders. The sorted data were sent for rolling-ball background subtraction and ACsN denoising[46] for image SNR enhancement. If the image has a size of 1024 × 900 pixels, it will then be padded to 1024 × 1024 pixels. For 3D reconstruction, we employed a graphic card (Titan RTX, Nvidia) to accelerate Richardson-Lucy deconvolution (RLD) (Supplementary Note 5). For phantom imaging, we used 30 iterations for the RLD of fluorescent microspheres. For biological samples, we used 50 to 80 iterations for RLD. Our current desktop can complete a single iteration with a 1024 pixels × 1024 pixels × 101 pixels

hybrid PSF within 0.2 s, and thus, a single 3D volume can be recovered within about 10 seconds[43] (Supplementary Note 12 and Supplementary Table 4). For volume visualization, ClearVolume[71] (version 1.4.2) was used to render 3D volume in all figures except for Fig. 2b–d and Fig. 4h, which is rendered using PyVista[72] (version 0.38.4) with customized code.

## Bead phantom preparation

We used 200 nm, 1 μm, 2 μm, and 4 μm fluorescent beads for phantom sample imaging. We mixed the four types of beads with an amount of 10 μL, 50 μL, 100 μL, and 200 μL, respectively, and diluted the solution to 3 mL with 1× PBS for experimental observation.

## Animal experiments

All animals used in this study were housed at the animal facility at Georgia Institute of Technology. Ai14 mice (age of 6–18 weeks) and C57BL/6J mice (age of 8–12 weeks) were housed with a room temperature range between 20 and 26 °C, humidity of 40–70%, and a semi-natural light cycle of 12:12 light-to-dark ratio. OT-I mice (age of 6–8 weeks) and C57BL/6 mice (age of 6–8 weeks) were housed with a room temperature range between 20 and 21.7 °C, the humidity of 30–55%, and the semi-natural light cycle of 12:12 light-to-dark ratio.

## HeLa cell culture and mitochondria and peroxisome two-color staining

HeLa cells (#93021013, Sigma-Aldrich) were cultured in Dulbecco's modified Eagle medium (DMEM) with 10% fetal bovine serum (FBS) and 1% Penicillin-Streptomycin (Pen-Strep) at 37 °C in a 5% $CO_2$ atmosphere. Before the imaging day, the cells were incubated in a pre-warmed (37 °C) mixed solution containing 3 mL modified DMEM and 60 μL Peroxisome-GFP. The GFP was expressed on the peroxisomes after 22 hours of incubation.

On the imaging day, 0.3 μL of 1 mM MitoTracker Deep Red FM stains were added to the growth medium. The cells were incubated for an additional 30 min. Then, the growth medium was removed, and the cells were washed twice using HBSS without phenol red. After HBSS was removed, 1.5 mL of trypsin-EDTA was added to the dish for 1 min, gently swirled, and removed. The cell dish was placed inside the incubator for 3 min to detach the cells. Once incubation was done, cells were resuspended into 3 mL of 4% PFA fixation buffer (16% PFA with PFA:PBS:ultrapure-water in a 1:2:1 ratio) in a 5 mL vial at room temperature for 12 min. Cells were concentrated by centrifuging for 6 min at 800 × g. Then, cells were resuspended into 3 mL of clear PBS. This washing step was repeated again, and cells were finally stored in 3 mL of PBS without phenol red for imaging.

## Mouse spleen and blood cell isolation and staining

C57BL/6J mice ($N$ = 3 mice/group) were sacrificed for spleen cells and blood cell collection. The spleen was minced and transferred to Eppendorf tubes containing 1× PBS. Next, it was filtered through a 70 μm mesh (Biologix Research Company 15-1070); 7 mL of PBS was added, and the cell suspensions were centrifuged at 800× g for 7 min. Spleen cells were subsequently resuspended in 1× PBS. Blood was collected through cardiac puncture. Blood cells were washed with 1× PBS and resuspended in 1× PBS for further imaging processing.

For spleen cells, 15 μL of wheat germ agglutinin (WGA) was added to each vial of spleen cells for 25 min incubation at 37 °C. Then, the solution was washed twice with 1.5 mL phosphate-buffered saline (PBS) in the centrifuge with 800× g for 6 min. 1.5 mL of 4% paraformaldehyde solution (PFA) was added to perform fixation at room temperature for 15 min. After fixation, the solution was washed twice with 1.5 mL PBS in the centrifuge with 800 × g for 6 min. 1.5 mL PBS with 5 mM EDTA was added to each vial before imaging.

For blood cells, 15 μL of WGA was added to each vial of blood cells for 25 min incubation at 37 °C. Then, the solution was washed twice

with 1.5 mL PBS in the centrifuge with 500 × g for 5 min. 1.5 mL of 4% PFA was added to perform fixation at room temperature for 15 min. After the fixation, the solution was washed twice with 1.5 mL PBS in the centrifuge with 500 × g for 5 min. 3 mL PBS with 5 mM EDTA was added to each vial before imaging.

## Mouse naïve T cell isolation and staining

OT-I mice (N = 1 mice/group) were sacrificed, and the spleens were mechanically digested into cell suspension, and CD8 + T cells were negatively purified from cell suspension with an untouched CD8 + T cell isolation kit. On the imaging day, 2.5 mL of T cells suspended in Rosewell Park Memorial Institute (RPMI) medium was transferred into a 35 mm FluoroDish. 12.5 µL of SYTO16 was added to the dish, and cells were incubated for 1 hour. At the 45-min time point, 2.5 µL of 1.2X HCS CellMask Deep Red staining solution was added. The 1.2X HCS Cell-Mask staining solution was prepared by adding 2.4 µL of the HCS CellMask stock solution (250 µg HCS CellMask Stain with 100 µL of Dimethyl sulfoxide) to 10 mL PBS. The staining of HCS CellMask and SYTO16 was completed at the same time. Then cells were transferred to a 5 mL vial and centrifuged (300× g, 16 min) to be collected. The cell pellet was resuspended using 1.8 mL of HBSS. The centrifuge-resuspending procedure was repeated twice. In the last round, cells were resuspended into 1.8 mL of the 4% PFA fixation buffer and centrifuged again (300 × g, 16 min). Finally, the supernatant was discarded carefully, and cells were resuspended into PBS for storage and flow cytometry imaging.

## Human-activated T-cell isolation and staining

Human immune cells (peripheral blood mononuclear cells) were isolated by density gradient centrifugation (Lymphoprep density gradient medium and SepMate-15mL tube). The cells were separated by a selection kit (EasySep Human CD3 Positive Selection Kit II). Dynabeads Human T-Activator was used at the ratio of 3:1 (bead-to-cell) to activate the T cells. With complete human T cell media (X-vivo 10 Serum-free Hematopoietic Cell Medium, 5% Human AB serum, 10 mM N-Acetyl-L-cysteine, and 55 µM 2-Mercaptoethanol), the mixture of activated cells was cultured and maintained with supplements (50 µg/mL recombinant human IL-2) at the concentration of $7 \times 10^5$ to $2 \times 10^6$ cells/mL. After day 7 of the culture, the mixture was diluted with Dynabeads at a ratio of 1:1 (bead-to-cell). On Day 9, human T cells were isolated at a concentration of between $7 \times 10^5$ and $2 \times 10^6$ cells/mL.

On the imaging day, 12.5 µL of SYTO16 was added to the dish, and cells were incubated for 1 hour. At the 45-min time point, 2.5 µL of 1.2X HCS CellMask Deep Red staining solution was added for another 15-min staining. Then cells were transferred to a 5 mL vial and centrifuged (300× g, 16 min) to be collected. The cell pellet was resuspended using 1.8 mL of HBSS. The centrifuge-resuspending procedure was repeated twice. In the last round, cells were resuspended into 1.8 mL of the 4% PFA fixation buffer and centrifuged again (300× g, 16 min). Finally, the supernatant was discarded carefully, and cells were resuspended into PBS for storage and flow cytometry imaging.

## Jurkat cell apoptosis induced by staurosporine (STS) treatment and staining

Jurkat T cells (#88042803, Sigma-Aldrich) were cultured in RPMI with 10% FBS and 1% Pen-Strep as a modified RPMI medium at 37 °C and in a 5% $CO_2$ environment. On the imaging day, 1 µM of STS was added to 4 cell dishes, incubating for 30, 60, 120, and 300 min, respectively, at 37 °C. The following procedures for preparing cells in the 4 dishes are the same. After the treatment, cells were centrifuged (500 × g, 6 min, 37 °C) and resuspended into the modified RPMI medium. After two rounds of centrifuge-washing, cells were resuspended into 6 mL of modified RPMI medium. For fluorescence labeling, 250 nM of SYTO16 green stains were added to the culture dish, and the cells were incubated for 1 hour. At the halfway of

incubation (30 min), 150 nM of MitoTracker Deep Red FM stains were added to the culture dish. After another 30 min, cells were centrifuged (500 × g, 6 min, 37 °C) and resuspended into 6 mL of the 4% PFA fixation buffer. Cells were fixed at room temperature for 12 min. After fixation, cells were centrifuged (800 × g, 6 min, room temperature) and resuspended into PBS twice. Finally, cells were stored in PBS for imaging flow cytometry.

## Nanoparticle formulation and characterization

20α-OH cholesterol lipid nanoparticle[58] was formulated using a microfluidic device as previously described[59]. Briefly, lipid nano-particle was created through the rapid mixing of aqueous and organic phases in a custom-made microfluidic device that uses syringes for each phase, with a 3:1 flow rate (aqueous to organic). Cre mRNA[73] was diluted in 10 mM citrate buffer. cKK-E12 was purchased from Oragnix Inc. (O-8744). C18PEG2K and 18:1 (Δ9-Cis) PE (DOPE) were diluted in 100% ethanol and purchased from Avanti Lipids. Citrate and ethanol phases were combined in a microfluidic device by syringes at a flow rate of 3:1. The diameter and poly-dispersity of the LNPs were measured using dynamic light scatter-ing (DLS). LNPs were diluted in sterile 1X PBS and analyzed. Particles were dialyzed in Slide-A-Lyzer G2 20 kD dialysis cassettes from Thermo Scientific, and the nanoparticle concentration was deter-mined using NanoDrop.

## Cell isolation and staining for Cre mRNA delivery experiments

Ai14 mice (N = 4 mice/group) was injected with 20α-OH cholesterol lipid nanoparticle at a total dose of 0.25 mg/kg nucleic acid. Mice were sacrificed, and cells were isolated 72 h after injection with LNPs. Mice were perfused with 20 mL of 1X PBS through the right atrium. The liver, spleen, and lung were isolated. The liver and lung were finely cut and then placed in a digestive enzyme solution with Collagenase Type I, Collagenase XI, and Hyaluronidase at 37 °C at 550 rpm for 45 min. The spleen was appropriately minced and placed in 1X PBS. The cell sus-pension was filtered through 70 µm mesh, 7 mL of PBS was added, and the cell suspensions were centrifuged at 800× g for 7 min. Next, Lung and liver cells were stained with anti-CD31 (1:200 dilution), and the spleen was stained with anti-CD45 (1:200 dilution). Prior to staining, FC receptors were blocked with TruStain FcX™ antibody (1:100 dilution in 1X PBS) to avoid non-specific binding. Next, the samples were kept at 4 °C for 45 min until the staining was complete. The samples were then washed with 1× PBS and transferred to Eppendorf tubes, resuspended in 1X PBS for further imaging processing.

## Chemicals and biological materials

The sources of the chemicals and biological materials used in the experiments, including company names and catalog numbers, are listed in Supplementary Table 3.

## Statistics and reproducibility

The fluorescence staining protocols were repeated at least twice for each experiment. During the data acquisition, samples were loaded and imaged by at least three independent imaging sessions for each experiment.

## Reporting summary

Further information on research design is available in the Nature Portfolio Reporting Summary linked to this article.

# Data availability

Source data are provided with this paper. The datasets generated and analyzed in the manuscript are available from https://doi.org/10.5281/zenodo.10471580. Additional datasets are available from the corre-sponding author upon request due to the large file size. Requests will be fulfilled within two weeks. Source data are provided with this paper.

## Code availability

The code package for the LFC system is available as Supplementary Software 1. The code is written in MATLAB 2021b and 2022a (MathWorks) and Python 3.9. The latest version of the software is available at https://github.com/ShuJiaLab/LFC.

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

## Acknowledgements

We acknowledge the support of the National Institutes of Health grants R01AI171892 (to G.A.K.), R01HL132019 and U01CA214354 (to C.Z.), R35GM124846 (to S.J.), the National Science Foundation grants EFMA1830941 and DBI2145235 (to S.J.), the National Science Foundation Graduate Research Fellowships DGE-2039655 (to C.Z.) and DGE-2039655 (to A.D.S.T.), and the National Institutes of Health Cell and Tissue Engineering Training Program T32GM145735 (to A.D.S.T.). We would like to thank Kevin Ferri, James Steinberg, Larissa Doudy, and Sandy Hsieh at Georgia Institute of Technology for their technical support.

## Author contributions

X.H. and S.J. conceived and designed the project. X.H., K.H., B.M., W.L., and C.Z. contributed to constructing the imaging flow cytometry system. A.R., H.K., and J.D. helped with mouse blood cells and liver, lung, and spleen cells after Cre mRNA delivery in Ai14 mice. S.E., Z.Y., K.L., and C.Z. helped with mouse naïve T cells and platelets. A.D.S.T. and G.A.K. helped with the human T-cell samples. X.H., K.H., and J.S. prepared and labeled samples. X.H. and K.H. performed imaging experiments and data analysis. X.H. developed image processing algorithms. G.A.K., C.Z., and J.D. contributed scientific insights. S.J. supervised the overall project. X.H., K.H., and S.J. wrote the manuscript with input from all authors.

## Competing interests

The authors declare no competing interests.
