## [Peer Review File · Nature Communications]

REVIEWER COMMENTS

Reviewer #1 (Remarks to the Author)

This manuscript presented a LFC (Laser Fluorescence Cytometry) system that integrated flow cytometry and fluorescence microscopy. Then the LFC system was applied to different scenarios. However, the results in the manuscript did not clearly demonstrate the necessity of the development of the system. Subcellular structures have been studied at 3D level using other imaging systems with much higher spatial resolution. More important, although the LFC system allowed for a rapid analysis, its spatial resolution was between 300 and 600 nm, which was too poor to provide useful information about 3D subcellular characteristics. The applications proposed by the authors did not show the advantages and necessity of the system. Due to this lack of novelty and significance of the results, the manuscript is therefore not recommended for publication in Nature Communications :

Major concerns:

(1) Fig. 3: The authors should compare the images of peroxisomes and mitochondria obtained by the LFC system with those obtained by other imaging systems (such as 3D-SIM) to illustrate the accuracy. In my opinion, the spatial resolution of both peroxisomes and mitochondria in Figure 3 is too poor to provide reliable information for quantitative analysis, especially for peroxisomes with size much smaller than 300 nm.

(2) Fig. 5: The images of mitochondria and nucleus obtained by the LFC system should be compared with those obtained by other imaging systems (such as 3D-SIM) to illustrate the accuracy. Compared to flow Cytometry alone, how did the unclear 3D visualization of individual cells obtained by the LFC system improve the results? What are the new conclusions?

(3) It is worth noting that although the acquisition speed of the LFC system was up to 5,750 cells per second, faster imaging speed may result in a worse signal-to-noise ratio and lower spatial resolution. There were no results in the manuscript to demonstrate the necessity and advantages of such a high acquisition speed.

(4) The X-Y resolution of mitochondria (Fig. 3) was approximately 560 nm, too poor to clearly identify mitochondria.

Minor concerns:

(1) Fig.2: What is the accuracy of identifying microbeads using the LFC system?

(2) Fig2i-k: What did the different colors represent in the scatter plots?

(3) The spatial resolution of Fig. 5i was obviously worse than that of Fig. 5f. Thus, the morphological changes in mitochondria and nuclei may be due to the decreased resolution.

Reviewer #2 (Remarks to the Author)

The manuscript describes the development of a flow cytometry-based 3D single cell image acquisition system. The 3D imaging is achieved using light field microscopy which can reconstruct a 3D image from a single image using a micro-lens array. The image is reconstructed computationally using deconvolution of the captured micro-lens image. The coupling of a light field imaging system with a micro fluidic channel system appears challenging and the authors have achieved an impressive throughput of cells and the image resolution looks to be an improvement on current imaging flow cytometry systems.

The development of Imaging Flow Cytometry systems is a highly active area with new devices and techniques used for imaging appearing constantly. The specification of this proposed system is loosely compared with other technologies such as three-dimensional localization microscopy, light sheet microscopy, confocal etc. The authors claim a major advantage of this system over current technologies is the improvement of the spatial image resolution. Although reference 22 - Weiss, L.E. et al. also claim sub-micron resolution so a detailed comparison of advantages of the technology described here would be desirable to allow the reader to decide on the advances for this system. Also, some improved versions and new technologies have appeared recently which warrant a discussion where this system stands in comparison e.g. Ugawa et.al. Biomed Opt.

Express 2022, Kumar et.al. Scientific Reports 2022, Gong et.al. Micromachines 2023. Some of these technologies include very similar experimental imaging exemplars and therefore a comparison with the latest systems is also important to assess the contribution made here.

The manuscript is well written, and the example imaging experiments are well defined and show the systems capabilities. However, one concern with the manuscript is the description of the deconvolution of the micro-lens image which is very brief, and I would like to see far more detail on the image construction especially on the process of determining the point spread function which is key to this procedure.

Reviewer #3 (Remarks to the Author)

In this paper, the authors manufactured a microfluidic setup based on a previously published work about light-field microscope (high-resolution Fourier light-field microscope, Hua, X. et al, Optica, 8(5), 614-620, 2021), to develop a new LFC system, and used it to perform flowing cell experiments. While this work expands the application range of light-field microscope, it does not appear to introduce significant advancements in optics or algorithms compared to the authors' previous work. Therefore, before making a further consideration, several major points need to be addressed first to show its importance in applications.

1. The authors should better demonstrate the advances of the LFC versus the previous high-resolution Fourier light-field microscope (Hua, X. et al, Optica, 8(5), 614-620, 2021), in terms of optics and algorithms. If the main contribution of this work is only the extended application to IFC experiments with the microfluidic devices, its suitability for publication in Nature Communications may be questionable.

2. I observed that the point spread function (PSF) depicted in Fig. 1c of this paper seems to be nearly identical to Fig. 1b of the previous paper (Hua, X. et al, Optica, 8(5), 614-620, 2021). It is important to clarify whether there are any changes in the optical parameters of the LFC system compared to the previous high-resolution Fourier light-field microscope. If there are changes, the authors should explain why these modifications were necessary to adapt to the imaging flow cytometry (IFC) case.

3. The authors used the stroboscopic illumination to reduce the motion blur. How about the distance does the sample move during the stroboscopic time? Is it less than the lateral resolution of the system? Please quantify it.

Furthermore, a tradeoff between signal-to-noise ratio and motion blur exists, where the shorter stroboscopic time reduces motion blur but may weaken the signal-to-noise ratio. Conducting additional experiments to quantitatively clarify the choice of these parameters is recommended.

4. From the raw LFC images (for example, Figs. 2a, 2e, 2f, 4g, 5a, 5b...), the signal-to-noise ratio appears to be enough for the subsequent deconvolution process. Therefore, it raises a question of whether the denoising algorithm is necessary at all times. It would be valuable to discuss the necessity of the denoising algorithm and examine whether there is any loss of resolution after denoising. I recommended that the authors could add an experimental comparison of LFC reconstruction results without and with pre-denoising, preferably at different noise levels.

5. The authors mentioned that the depth of field of LFC is 3-4 μm , which seems to be sufficient only for the observation of cultured adherent cells. However, for flowing cells, the depth-of-field range may not cover the entire cell due to the diameter of the microfluidic channel. Will this become a problem for practical application? How can it be solved? I think it is quite necessary for the 3D imaging of flowing cytometry. Or maybe people can directly use some methods with extended depth of field for 2D imaging without the requirement for the axial information? The necessity of axial resolution has not been demonstrated.

Overall, addressing these major points will strengthen the manuscript and help in evaluating its suitability for publication in Nature Communications.

RESPONSE TO REVIEWERS' COMMENTS

We thank the Reviewers for thoroughly examining the manuscript and providing constructive comments. Here, we submit a point-by-point response letter in which we have provided detailed responses to each comment from the Reviewers and made corresponding revisions in our revised manuscript as presented in the following.

Response to Reviewer #1

This manuscript presented a LFC (Laser Fluorescence Cytometry) system that integrated flow cytometry and fluorescence microscopy. Then the LFC system was applied to different scenarios. However, the results in the manuscript did not clearly demonstrate the necessity of the development of the system. Subcellular structures have been studied at 3D level using other imaging systems with much higher spatial resolution. More important, although the LFC system allowed for a rapid analysis, its spatial resolution was between 300 and 600 nm, which was too poor to provide useful information about 3D subcellular characteristics. The applications proposed by the authors did not show the advantages and necessity of the system. Due to this lack of novelty and significance of the results, the manuscript is therefore not recommended for publication in Nature Communications:

Response: We appreciate the thoughtful comments from the Reviewer, which have guided us in clarifying the novelty and significance of our research. In this letter, we have provided detailed responses to each comment raised by the Reviewer. We would like to clarify that imaging flow cytometry (IFC) techniques offer unprecedented throughput compared to conventional optical microscopy systems, which are significant and critically demanded for both basic and translational single-cell analysis at the population level¹. In this context, **light-field flow cytometry (LFC)** in this work presents novel instrumental and computational development, representing a leading 3D-IFC system that outperforms existing IFC techniques in terms of 3D imaging capability, spatial resolution, and imaging throughput (**Supplementary Table 2, Supplementary Figure 24 in Supplementary Note 11**). It should be mentioned that while presenting the highest throughput among current 3D IFC techniques, the spatial resolution of 300-600 nm obtained by LFC has already achieved the diffraction limit of high-resolution optical microscopy and IFC. Although super-resolution techniques such as SIM can attain a sub-diffraction-limited resolution, they remain primarily incompatible with the flow setting due to their sequential (rather than snapshot and volumetric as in LFC) acquisition scheme (**Figs. R1 and R2 in Responses to Major Concerns 1 and 2, Supplementary Notes 8 and 10**). As a result, the available super-resolution strategies still rely on conventional platforms to trap and acquire static super-resolution images, unable to preserve the throughput^{2,3}. We have pioneered a technique termed optofluidic scanning microscopy (OSM)⁴ based on image-scanning microscopy (a variant of SIM) to achieve the first super-resolution acquisition compatible with flowing cells, but this method was realized at a significantly lower throughput (a few cells per sec, vs. thousands of cells per sec for LFC). Therefore, we believe LFC features a novel and significant advance for highly desirable cytometric imaging techniques for a broad range of cell biological studies.

Major concerns:

(1) Fig. 3: The authors should compare the images of peroxisomes and mitochondria obtained by the LFC system with those obtained by other imaging systems (such as 3D-SIM) to illustrate the accuracy. In my opinion, the spatial resolution of both peroxisomes and mitochondria in Figure 3 is too poor to provide reliable information for quantitative analysis, especially for peroxisomes with size much smaller than 300 nm.

Response: First, as suggested by the Reviewer, we performed additional experiments and analysis by imaging mitochondria and peroxisomes in HeLa cells across multiple platforms. In particular, we compared the results generated by light-field microscopy (**Nikon Eclipse Ti2U**; OBJ: 100×/1.45NA), wide-field microscopy (the same **Nikon Eclipse Ti2U** setup as for LFC), and, as mentioned by the Reviewer, commercial 3D-SIM microscopy (**Zeiss LSM 780** with **Zeiss ELYRA PS.1**; OBJ: 100×/1.46NA). Also, we used cultured HeLa cells and ensured the quantitative comparison of the same cells across multiple modalities. As seen in **Figure R1** (see next

page and revised **Supplementary Note 8**), the 3D light-field results of subcellular structures displayed a high consistency compared with the scanning wide-field stacks and 3D super-resolution results. Furthermore, we employed the 3D structural similarity index measure (SSIM) to quantitatively verify the accuracy of the results. 3D-SSIMs of mitochondria and peroxisomes exhibited (0.83, 0.95; wide-field) and (0.78, 0.82; SIM) of the light-field images in comparison with the wide-field and SIM results, respectively. In addition to our demonstration, we would also like to clarify that the accuracy and fidelity of light-field microscopy techniques have been demonstrated utilizing various sample types and imaging conditions. Also, to address the flow setting in this study, we used hybrid point-spread functions (hPSFs)^{5,6}, considering the spherical aberration caused by the depth in the flow and any actual experimental misalignments and imperfections. In summary, we expect these results and elaborations to clarify the high accuracy of 3D structural retrieval of light-field images consistent with the wide-field and super-resolution SIM images.

Next, we would like to clarify the spatial resolution of 300-600 nm obtained by LFC. In fact, these values have already achieved the diffraction limit of high-resolution optical microscopy for cell biology, representing the state-of-the-art performance of imaging flow cytometry (IFC) (see below **Table R1** and revised **Supplementary Note 11**). Major high-resolution imaging cytometers can only capture 2D images. Indeed, super-resolution techniques such as SIM can attain a sub-diffraction-limited resolution but remain largely incompatible with the flow setting due to their sequential (rather than snapshot and volumetric) acquisition scheme. As a result, the available super-resolution strategies still rely on conventional platforms to trap and acquire static super-resolution images, unable to preserve the throughput^{2,3}. As another example, we have also developed a technique termed optofluidic scanning microscopy (OSM)⁴ based on image-scanning microscopy (a variant of SIM) to achieve the first super-resolution acquisition of flow cells, but the method was still realized at a significantly reduced throughput (a few cells per second)⁴. In contrast, LFC presents about three orders of magnitude of improvement in the throughput and diffraction-limited 3D resolution, which could be the main desirable and long-seeking feature for researchers using IFC. Regarding the further improvement of the 3D resolution, we have recently reported a computational strategy (Han, 2022)⁷ based on the radially of the light-field images to achieve a resolution doubling comparable to SIM (see **Figure R5** and revised **Supplementary Note 5**). Lastly, we would like to note that various peroxisomal and mitochondrial studies have already been conducted using fluorescence microscopy featuring similar or even lower resolution to identify and localize subcellular peroxisomal and mitochondrial processes and structures (for example, *Supp. Refs. 8,9*). In this sense, we do not fully agree with the Reviewer's comment that "*the spatial resolution ... is too poor to provide reliable information for quantitative analysis*". In summary, we expect our results and elaborations to demonstrate that LFC is accurate, state-of-the-art, and suitable for a broad range of cell biological studies.

Figure R1. 3D-rendered (leftmost column) and axial slices at various depths (other columns) of mitochondria (a, b, c) and peroxisome (d, e, f) of HeLa cells, taken by LFC (a, d), scanning wide-field microscopy (b, e), and 3D SIM (c, f). The boxed regions were zoomed in for better visualization and comparison. The green arrows mark the corresponding structures displayed using each imaging method. The 3D structural similarity index measure (SSIM) values were shown in the wide-field and SIM images. Scale bars: 10 μm .

(2) Fig. 5: The images of mitochondria and nuclei obtained by the LFC system should be compared with those obtained by other imaging systems (such as 3D-SIM) to illustrate the accuracy. Compared to flow Cytometry alone, how did the unclear 3D visualization of individual cells obtained by the LFC system improve the results? What are the new conclusions?

Response: In light of this comment on **Figure 5**, we performed additional imaging experiments of staurosporine (STS) induced Jurkat cells, across multiple platforms. Similar to the previous response, we utilized light-

field microscopy (**Nikon Eclipse Ti2U**; OBJ: 100×/1.45NA), wide-field microscopy (the same **Nikon Eclipse Ti2U** setup as for LFC), and commercial 3D-SIM microscopy (**Zeiss LSM 780** with **Zeiss ELYRA PS.1**; OBJ: 100×/1.46NA). Also, using a lab-derived protocol (see **Supplementary Note 10**) that treated the coverslip, we immobilized Jurkat cells in the dish to ensure the quantitative comparison of the same cells across multiple modalities. As seen in **Figure R2**, we observed 3D-SSIM values of mitochondria at 0.87 (0 min after STS treatment), 0.87 (30 min), and 0.89 (120 min) between the wide-field and light-field results. Similar high consistency is also verified between the SIM and light-field results, with 3D-SSIM values of 0.89 (0 min), 0.94 (30 min), and 0.97 (120 min). Consistent quantitative measurement (0.80, 0.81, and 0.77 for respective treatment times) was obtained for nucleus imaging using light-field and wide-field microscopy. We expect these supportive results, combining our response to comment #1, to verify the high accuracy of 3D structural retrieval of light-field images consistent with the wide-field and super-resolution SIM images (see **Supplementary Table 4**).

Furthermore, we would like to emphasize the comment “compared to flow cytometry alone”, as raised by the Reviewer. LFC, as a state-of-the-art IFC technique, offers diffraction-limited resolution in all three dimensions with a high throughput and multicolor ability. The visualization enabled by LFC offers highly desirable 3D morphological characteristics, advancing traditional flow cytometers that lack morphological details. Specifically, in contrast to traditional cytometry relying primarily on single-cell fluorescence and scattering, the LFC system can obtain 3D images of single-cell populations, revealing subcellular features, such as size, shape, biomarker intensity, physiological state, and other morphological and biochemical characteristics¹⁰. This enhanced information significantly advances cell biological discovery, allowing for a more comprehensive understanding of populational heterogeneity beyond what is achievable with flow cytometry alone. As an IFC platform, this light-field cytometric approach will open new possibilities in various research areas, including basic cell biology, disease diagnostics, and therapeutics.

Figure R2. 3D rendering of reconstructed light-field volumes, wide-field z-scanning stacks, and SIM z-scanning stacks of Jurkat cells with staurosporine (STS) treatment. (a, c, e) Jurkat cell mitochondria with 0-min (a), 30-min (c), and 120-min (e) STS treatment. (b, d, f) Jurkat cell nuclei with 0-min (a), 30-min (c), and 120-min (e) STS treatment.

(3) It is worth noting that although the acquisition speed of the LFC system was up to 5,750 cells per second, faster imaging speed may result in a worse signal-to-noise ratio and lower spatial resolution. There were no results in the manuscript to demonstrate the necessity and advantages of such a high acquisition speed.

Response: Indeed, we agree with the Reviewer that a higher throughput typically poses IFC techniques a lowered SNR (and thereby compromised image quality) due to a correspondingly shorter exposure (fewer photons captured) of individual objects. To address this challenge, in this work, we used a lab-written algorithm (termed ACsN¹¹), which has been verified with its effectiveness for restoring low-SNR light-field images in various conditions^{5-7,12}. As a result, in our work, we observed that even with 5- μ s stroboscopic illumination, the raw fluorescent light-field signals can still be restored with ACsN and recovered into 3D images (**Figure R3** and revised **Supplementary Note 6**). It should be mentioned that novel hardware such as VIFFI¹³ or FIRE¹⁴ has also been employed in other IFC techniques to achieve high SNR in spite of the cost of instrumental complexity.

Furthermore, importantly, we would like to clarify that the throughput is one of the essential and most sought-after characteristics of IFC, or optofluidics in general, as microscopy-based imaging systems are demanded for rapid, comprehensive, and statistically robust analysis of cell populations. In this sense, for example, the gold-standard commercial IFC systems can achieve a throughput of approximately 2,000 cells/s using 40 \times magnification¹⁵. In practice, a higher throughput of IFC allows us to efficiently acquire large datasets, identify and sort rare cell populations, conduct multi-parameter analysis, and generate statistical significance. In this context, as shown in revised **Supplementary Note 11**, we have included **Supplementary Table 2** for comparing LFC and other state-of-the-art IFC techniques in terms of spatial resolution and imaging throughput. In the corresponding **Figure R4** (revised **Supplementary Figure 24**), we provided a clear view of the states of LFC and other 2D and 3D IFC techniques. The plot shows that LFC possesses an advantageous diffraction-limited spatial resolution and the highest analytical throughput among all 3D IFC techniques.

Figure R3. (a) The raw light-field image of Jurkat cell nucleus with dual snapshots within one frame under 5- μ s stroboscopic illumination. (b) The 3D reconstructed volume of (a). (c) The 2D focal stack image of (b). (d) The light-field image of Jurkat cell nucleus in (a) with ACsN. (e) The 3D reconstructed volume of (d). (f) The 2D focal stack image of (e), in which nucleus details were better resolved. Scale bars: 10 μ m (a, d), 1 μ m (c, f).

Figure R4. State-of-the-art IFC techniques. Red diamond: LFC; Round black dots, 3D IFC techniques. Gray crosses, 2D IFC techniques. Miura, 2018 showed 2D results and claimed the system is capable of 3D imaging.

(4) The X-Y resolution of mitochondria (Fig. 3) was approximately 560 nm, too poor to clearly identify mitochondria.

Response: We appreciate the opportunity to clarify this point, which is related to our response to comment #1: *(i)* Spatial resolution: We would like to emphasize that the 300-600 nm spatial resolution achieved by LFC already approaches the diffraction limit for high-resolution optical microscopy in cell biology, representing the current state-of-the-art in IFC (**Table R1** and **Supplementary Note 11**); *(ii)* Super-resolution techniques such as SIM can provide higher resolution but are largely incompatible with flow settings due to their sequential acquisition scheme; *(iii)* In contrast to our recent 2D super-resolution IFC (OSM)⁴, LFC offers significantly improved throughput of around three orders of magnitude while maintaining diffraction-limited 3D resolution; *(iv)* We would like to address that a wide range of studies on mitochondria have been successfully conducted using fluorescence microscopy with similar or even lower resolution (for example, **Supplementary Refs** ¹⁶⁻¹⁸) while LFC offers an additional dimension to the cell population; *(v)* Lastly, we recently proposed 3D resolution doubling without compromising the throughput using the radially of the light-field images (i.e., *radFLFM*, Han, 2022)⁷ (see **Figure R5**, revised **Supplementary Note 5** and revised **Main Text Discussion**). In summary, we respectfully disagree with the assertion that the achieved resolution is insufficient for reliable quantitative analysis of mitochondria and offers further improvement strategy in the manuscript for readers' information.

Figure R5. (a) Raw light-field images of mitochondria in flowing HeLa cells. (b) The 3D reconstruction of (a). The depth information was color-coded according to the color scale bar. (c) Corresponding light-field image in (a) using ACsN and radiality analysis. (d) The corresponding 3D reconstruction of (c). Scale bars: 10 μm (a, c), 5 μm (b, d).

Minor concerns:

(5) Fig.2: What is the accuracy of identifying microbeads using the LFC system?

Response: In light of this comment, we performed extra experiments with fluorescent microspheres of the four sizes used in original **Figure 2**, i.e., 200 nm, 1 μm , 2 μm , and 4 μm . We used wide-field scanning images as the ground truth and compared the light-field imaging results (see **Figure R6** and **Supplementary Figure 4**). Same as in our experiments to address the previous comments on the accuracy, we ensured imaging the same beads for quantitative analysis using both wide-field and LFC systems. As shown in **Figure R6 (a, d, g)**, the 3D details in the orthogonal views evidenced the consistency between the reconstructed light-field volumes and the wide-field scanning stacks. We also measured the profiles of the four types of beads in both light-field and wide-field images (for statistical significance, we used 50-100 beads for each group). The statistics were shown in **Figure R6 (b, c, e, f, h, i)** and revealed that the measurements conducted with both imaging methods provided consistent profiles for bead sizes of 1 μm , 2 μm , and 4 μm . Notably, for 200-nm beads, because the bead profiles are slightly below the spatial resolution (or the diffraction limit) of both methods. As a result, the measured values agreed with the predicted values derived by the convolution of the spatial resolution of each imaging system with the actual bead sizes. Moreover, because LFC can capture 3D images, we measured the bead volumes V and calculated the diameters using $D = \sqrt[3]{6V/\pi}$, showing consistent ~ 500 nm resolving ability with those reported in **Figure 2f-k**.

Figure R6. (a,d,g) Reconstructed light-field (LF) scanned volumes (left columns) and wide-field (WF) volumes (right columns) of 200-nm (top row), 1- μ m (second row), 2- μ m (third row), and 4- μ m (bottom row) beads peaked at 680-nm (a), 599-nm (d), and 516-nm (g) fluorescent emission. (b,c,e,f,h,i) Histograms of bead diameters measured using light-field images (b,e,h) and wide-field images (c,f,i) of peak spectra at 680 nm (b,c), 599 nm (e,f), and 516 nm (h,i). Scale bars: 10 μ m.

(6) Fig2i-k: What did the different colors represent in the scatter plots?

Response: We apologize for the confusion. The color gradient in the scatter plots in **Figure 2i-k** serves to visualize the density distribution of the beads based on their respective volumes and intensities. In light of this comment, we have added elaborations in the **Figure 2 caption** for readers' better information.

(7) The spatial resolution of Fig. 5i was obviously worse than that of Fig. 5f. Thus, the morphological changes in mitochondria and nuclei may be due to the decreased resolution.

Response: First, we would like to confirm that images under different STS treatments in **Figure 5** underwent the same image processing and 3D reconstruction. The quality of these 3D reconstructed images may vary due to the structure density and the depth position with respect to the effective volume acquisition. In addition, in our experimental procedure, we normally acquired more than 30 cells in each group for the statistical analysis in **Figure 5m-q**. Here, to clarify the consistency of resolution at different STS conditions, we showed below another 15 cells acquired in the 300-min STS-treatment group (**Figure R7**). The results demonstrated consistent resolution achieved in two-color imaging compared to other STS groups.

Figure R7. Two-color imaging of mitochondria and nucleus in 15 exemplary flowing Jurkat T cells with 300-min STS treatment times. The three panels show 3D rendering on the top of each panel and corresponding focal stack images at the bottom of each panel. A majority of results displayed a consistent resolution of the subcellular structures compared to those of other treatment times in **Figure 5**. Scale bars: 5 μ m.

Reviewer #2

The manuscript describes the development of a flow cytometry-based 3D single cell image acquisition system. The 3D imaging is achieved using light field microscopy which can reconstruct a 3D image from a single image using a micro-lens array. The image is reconstructed computationally using deconvolution of the captured micro-lens image. The coupling of a light field imaging system with a micro fluidic channel system appears challenging and the authors have achieved an impressive throughput of cells and the image resolution looks to be an improvement on current imaging flow cytometry systems.

The development of Imaging Flow Cytometry systems is a highly active area with new devices and techniques used for imaging appearing constantly. The specification of this proposed system is loosely compared with other technologies such as three-dimensional localization microscopy, light sheet microscopy, confocal etc. The authors claim a major advantage of this system over current technologies is the improvement of the spatial image resolution. Although reference 22 - Weiss, L.E. et al. also claim sub-micron resolution so a detailed comparison of advantages of the technology described here would be desirable to allow the reader to decide on the advances for this system. Also, some improved versions and new technologies have appeared recently which warrant a discussion where this system stands in comparison e.g. Ugawa et.al. *Biomed Opt. Express* 2022, Kumar et.al. *Scientific Reports* 2022, Gong et.al. *Micromachines* 2023. Some of these technologies include very similar experimental imaging exemplars and therefore a comparison with the latest systems is also important to assess the contribution made here.

The manuscript is well written, and the example imaging experiments are well defined and show the systems capabilities. However, one concern with the manuscript is the description of the deconvolution of the micro-lens image which is very brief, and I would like to see far more detail on the image construction especially on the process of determining the point spread function which is key to this procedure.

Response: We thank the Reviewer for the positive comments. First of all, we agree with the Reviewer that a detailed comparison will provide readers with better information on the state-of-the-art imaging flow cytometry (IFC) techniques. In addition, as mentioned by the Reviewer, we did not compare our proposed light-field flow cytometry (LFC) directly with existing 3D microscopy as they are not typically utilized (and may not be directly compatible) for cytometric imaging. However, we have cited the latest representative IFC works relying on relevant imaging strategies, including localization (Ref 22. Weiss, 2020), light-sheet (Ref 20. Gualda, 2017), and confocal (Ref 21. Quint, 2017) based IFC methods. As Ref 22 (Weiss 2020) pointed out by the Reviewer, we would like to clarify that precise localization of punctate flowing objects in a microfluidic environment has been reported. However, the localization-based approach requires a low emitter density to recognize individual flowing objects, incapable of capturing complex cellular structures. As a result, current strategies still rely on conventional platforms to trap and acquire super-resolution cell images^{2,3}, unable to preserve throughput and less synergistic for many live-cell IFC applications. In fact, we pioneered a technique termed optofluidic scanning microscopy (OSM⁴) to achieve the first super-resolution acquisition compatible with flowing cells, but this method was realized at a significantly lower throughput (a few cells per sec, vs. thousands of cells per sec for LFC). In light of this comment, for readers' better information, we have included a more thorough comparison among state-of-the-art IFC techniques (**Table R1**, **Figure R4**, and revised **Supplementary Note 11**). We have included the references mentioned by the Reviewer. Here we attached **Table R1** below for the Reviewer's convenience information.

Lastly, we appreciate the Reviewer's comment on the clarity of the processing framework. In light of this comment, we have included a detailed description of the vectorial Debye model utilized for the reconstruction in revised **Supplementary Note 4**. In brief, the process can be realized as the convolution between the volume of isotropic emitters in the object space and the PSF of the system. Therefore, the reconstruction becomes an

inverse problem of retrieving the radiant intensity at each point of the 3D object volume with the camera image. This process is a modified deconvolution algorithm based on the Richardson-Lucy iteration scheme. We have derived the theoretical models and formulas in revised **Supplementary Note 5**. Specifically, the spatial positions of the elemental images of the numerical PSF were adjusted based on the experimental PSF results (we called it a hybrid PSF, or hPSF). This strategy compensates for any instrumental alignment and settings between the theoretical model and the actual optical system (**Figure R8**). Moreover, such hPSF improves the SNR and image pixel precision (unsigned integer values by camera vs. double precision by simulation/hybrid PSF), thereby enhancing the accuracy of the 3D retrieval.

Figure R8. (a) Experimental PSF composed of three elemental light-field images. (b) PSF calibration between experimental (magenta) and simulated (green) PSFs. (c) Hybrid PSF (hPSF) after calibration. Scale bars: 10 μm (a, c), 5 μm (b).

Table R1. Comparison of the state-of-the-art IFC techniques

References	3D	Objective (M, NA)	Resolution (nm)	Throughput (objects/sec)	Imaging Instrument
LFC – this work	yes	100 \times , 1.45	300-600	5,000-10,000	Epi-fluorescence
Quint, et al., 2017 ¹⁹	yes	60 \times , 1.2	200-300	75-150	Tilted stage, Confocal tomography
Merola, et al., 2017 ²⁰	yes	60 \times , 1.2	360	2-3	Digital holography, tomography Not fluorescence IFC
Fan, et al., 2021 ¹⁶	yes	40 \times , 0.8	1,000	<10	Lattice light-sheet microscopy
Ugawa, et al., 2022	yes	20 \times , 0.75	880	1200	Strobe light-sheet imaging
Kleiber, et al., 2020 ²¹	yes	20 \times , 0.42	700-900	350	3D focusing tomography
Kumar, et al., 2022 ¹⁷ (VFC/iLIFE)	yes	20 \times , 0.4	1,786	10-20	Light-sheet microscopy
Han, et al., 2019 ²²	yes	10 \times , 0.28	2,000	500	Scanning light-sheet illumination
Zhang, et al., 2022 ²³	yes	10 \times , 0.28	10,000	500-1,000	Scanning light-sheet illumination
Miura, et al., 2018 ²⁴	double	20 \times , 0.75	1,000	10,000	Light-sheet microscopy
Nitta, et al., 2018 ²⁵ (IACS)	no	60 \times , 1.4	2,000	100	FDM confocal microscopy
Holzner, et al., 2021 ²⁶	no	40 \times , 0.75 20 \times , 0.5 10 \times , 0.5	500-1,000	5,350 (40 \times) 10,900 (20 \times) 20,500 (15 \times) 61,000 (10 \times)	Light-sheet illumination

Mikami, et al., 2020 ¹³ (VIFFI)	no	20×, 0.75	700	10,000	Polygon scanner light-sheet illumination
Isozaki, et al., 2020 ²⁷ (iIACS2.0)	no	20×, 0.75	700	2,000	Polygon scanner
Munoz, et al., 2018 ²⁸	no	20×, 0.45	700-900	7,000-23,000	Beat-frequency multiplexing
ThermoFisher Scientific (Attune CytPix)	no	20×, 0.45	800	6,000	Commercial system
Rane, et al., 2017 ²⁹	no	10×, 0.5 20×, 0.45	700-800	50,000- 100,000	Multichannel chip
Gong, et al., 2023	no	20×, 0.4	390	10,000	Anti-diffraction light sheet illumination
Ota, et al., 2018 ³⁰ (FiCS)	no	20×	1,000	10,000	Static random light structure Structured illumination/detection
Schraivogel, et al., 2022 ¹⁸ (ICS)	no	10×, 0.3	1,550	15,000	Radiofrequency-tagged emission (FIRE)
George, et al., 2004 ³¹ (Amnis ImageStream)	no	--, 0.75	500-1,000	1,200 (60×) 2,000 (40×)	Commercial system
Goda, et al., 2012 ³² (STEAM)	no	--, 0.65	1,400	100,000	Serial time-encoded amplified microscopy

Reviewer #3

In this paper, the authors manufactured a microfluidic setup based on a previously published work about light-field microscope (high-resolution Fourier light-field microscope, Hua, X. et al, *Optica*, 8(5), 614-620, 2021), to develop a new LFC system, and used it to perform flowing cell experiments. While this work expands the application range of light-field microscope, it does not appear to introduce significant advancements in optics or algorithms compared to the authors' previous work. Therefore, before making a further consideration, several major points need to be addressed first to show its importance in applications.

1. The authors should better demonstrate the advances of the LFC versus the previous high-resolution Fourier light-field microscope (Hua, X. et al, *Optica*, 8(5), 614-620, 2021), in terms of optics and algorithms. If the main contribution of this work is only the extended application to IFC experiments with the microfluidic devices, its suitability for publication in *Nature Communications* may be questionable.

Response: We thank the Reviewer for offering constructive comments and appreciate the opportunity to elucidate the significance of our work and address the concerns raised, particularly in relation to our prior research on high-resolution Fourier light-field microscopy (Hua, *Optica*, 2021). It is unequivocal that imaging flow cytometry (IFC) has transformed single-cell analysis, enabling comprehensive evaluations of cellular morphology and dynamic processes on a population scale. However, extant IFC methodologies are predominantly confined to capturing 2D images. Current strategies for 3D data acquisition in IFC—whether scanning or sequential [e.g., localization (Ref 22. Weiss, 2020), confocal (Ref 21. Quint, 2017), or light-sheet (Ref 20. Gualda, 2017)]—have been suboptimal for flow settings, often compromising key performance metrics like throughput and spatial resolution. Furthermore, integrating these 3D imaging techniques introduces additional layers of instrumental complexity and cost, limiting the widespread adoption of these advanced methods.

In light of these considerations, light-field microscopy emerges as an optimal candidate for 3D IFC. It distinguishes itself through its volumetric, snapshot, and high-resolution features, achieving the 3D acquisition of flowing cells without sacrificing either imaging or microfluidic performance. Moreover, light-field microscopy can be readily implemented using standard epi-fluorescence platforms, thus enhancing its accessibility and potential for broader research applications. Within this framework, light-field flow cytometry (LFC) transcends mere application extension of similar optical principles. Specifically, as detailed in the work, the system integrates innovations in optics, microfluidics, instrumentation, and computation to constitute a pivotal and highly desirable technological invention of 3D IFC. Without any of these innovations, it would be impossible to achieve the high-resolution, high-throughput 3D imaging capability for the LFC system. This innovative approach exerts a substantial influence on a diverse array of cell biological analyses. The comparative performance metrics of LFC, relative to existing IFC methodologies, have been systematically detailed in **Figure R4** and further elaborated in the revised **Supplementary Note 11**. In summary, we expect this response to clarify the novelty and significance of LFC and, thus, its suitability for publication in *Nature Communications*.

2. I observed that the point spread function (PSF) depicted in Fig. 1c of this paper seems to be nearly identical to Fig.1b of the previous paper (Hua, X. et al, *Optica*, 8(5), 614-620, 2021). It is important to clarify whether there are any changes in the optical parameters of the LFC system compared to the previous high-resolution Fourier light-field microscope. If there are changes, the authors should explain why these modifications were necessary to adapt to the imaging flow cytometry (IFC) case.

Response: We apologize for the confusion. Indeed, the PSF depicted in **Figure 1** of this paper appears similar to Figure 1b in the previous work. We should clarify that the experimental PSFs were acquired under consistent conditions, as the Fourier lens and MLA in LFC have been optimized for Fourier light-field image acquisition (**Supplementary Note 1**). However, computationally, in the subsequent 3D retrieval, the two works underwent

different procedures to generate the hybrid PSFs (hPSFs^{5,6}) to address disparate imaging conditions. In brief, hPSFs utilize the spatial positional information of the experimental PSFs while replacing the intensity profiles with the numerically simulated PSFs. This strategy (i) improves the SNR and image pixel precision (unsigned integer values by camera vs. double precision by simulation/hybrid PSF), thereby enhancing the accuracy of the 3D retrieval, while (ii) compensates for any instrumental alignment issues and deviations between the theoretical model and the actual optical system (**Figure R8** and revised **Supplementary Note 5**).

As elaborated in **Supplementary Note 4**, the actual PSF profiles are determined by variables including the numerical aperture of the objective lens, refractive-index mismatch (RIM), which can feasibly be addressed in the numerical PSFs and thus the hPSFs (**Eq. S6**). In our earlier work (Hua, *Optica*, 2021), the RIM remained negligible for imaging samples placed on the surface of the culture dish (near-zero normal focusing position (NFP)). As a result, the corresponding hPSFs generated in the previous work could largely ignore the RIM effects. In contrast, in the flow setting of cytometric imaging, cells are suspended in a fluid medium and traverse through a microfluidic channel. This alteration in sample conditions induces a significant distance between the cells and the channel bottom (5~10 μm), resulting in a *non-negligible RIM effect*. In this sense, we considered RIM factors in the optical model to ensure the hPSFs meet the actual microfluidic condition for reliable and accurate 3D reconstruction. We should mention that although the hPSF shown in **Figure 1d** looks similar to previous high-resolution light-field microscopy, they were actually distinct as adjusted for different experimental states (**Figure R9**). To clarify this confusion, we have elaborated it explicitly in the **Figure 1 caption** and added the hPSFs used in **Supplementary Figure 15** in the revised **Supplementary Note 5**.

Figure R9. (a) Hybrid PSF (hPSF) used in LFC. (b) hPSF used in previous work. (c) The subtraction of (a) and (b). (d) hPSF with medium refractive index (RI) 1.33. Scale bars: 10 μm .

3. The authors used the stroboscopic illumination to reduce the motion blur. How about the distance does the sample move during the stroboscopic time? Is it less than the lateral resolution of the system? Please quantify it. Furthermore, a tradeoff between signal-to-noise ratio and motion blur exists, where the shorter stroboscopic time reduces motion blur but may weaken the signal-to-noise ratio. Conducting additional experiments to quantitatively clarify the choice of these parameters is recommended.

Response: We appreciate the Reviewer's attention to the details concerning stroboscopic illumination and motion blur. The displacement \mathbf{d} of a sample during the stroboscopic illumination time T_{eff} can be quantified as $\mathbf{d} = \mathbf{v} \times T_{eff}$, where \mathbf{v} is the flow speed of the sample. To suppress motion blur, it is essential that this displacement \mathbf{d} be less than the lateral resolution of our LFC system at 400-600 nm. For experiments with $T_{eff} = 100 \mu\text{s}$, we constrained the sample speed to approximately 3 mm per sec, below the maximum allowable speed of 4-6 mm per sec, derived using $\mathbf{d} = \mathbf{v} \times T_{eff}$ based on the resolution. Similarly, for experiments with T_{eff} down to 5 μs , the flow speed was set to approximately 115 mm per sec, approaching the maximum allowable speed of 120 mm

per sec. As seen, under both conditions, the sample displacement captured within one stroboscopic frame remained shorter than the lateral resolution of LFC, thereby mitigating motion blur. In light of this comment, detailed quantitative derivation and elaboration have been incorporated into **Supplementary Note 3**. This revised version eliminates redundancy and streamlines the explanation, making it easier for the Reviewer to understand how you have addressed the concern about motion blur.

Furthermore, we agree that a shorter stroboscopic time reduces motion blur but may weaken the signal-to-noise ratio (SNR), owing to the reduced photon count during the effective exposure time. In this work, to address this trade-off, we have implemented our lab-written algorithms, including both background rejection and ACsN¹¹. While ACsN has previously been validated for its efficacy in restoring low-SNR light-field images^{5-7,12}, its deployment in flow setting has not been demonstrated. In addition, we have also rigorously optimized fluorescent staining protocols, selecting dyes and proteins through multiple rounds of testing to ensure robust results. As a result, our experimental data corroborate that even when utilizing stroboscopic illumination periods as brief as 5- μ s, our approach is proficient at reliably restoring raw fluorescent light-field signals for precise 3D image reconstruction (e.g., **Figure R3** and **Supplementary Figure 18**). It should be mentioned that efficient illumination and denoising have been primarily employed for recent high-throughput 2D IFC techniques, typically utilizing stroboscopic illumination durations of 10-20 μ s^{15,29}. It is also worth noting that hardware solutions such as VIFFI¹³ or FIRE¹⁴ have been proposed by high-throughput 2D IFC techniques to recover a high SNR, however, at the expense of increased instrumental complexity. Furthermore, as requested by the Reviewer, we have conducted additional experiments with various stroboscopic illumination periods. These experiments were accompanied by a quantitative analysis evaluating the SNR and image quality, both with and without the implementation of ACsN denoising (**Figure R10** and newly added **Supplementary Figure 19**). Our results substantiate that our strategy significantly elevates the quality of reconstructed images. This approach provides compelling evidence supporting the optimized parameters we have employed in the LFC system, effectively balancing robust SNR with minimal motion blur.

Figure R9. (a-d) Light-field images of the nucleus of flowing Jurkat cells with 5-ms (a), 2.7-ms (b), 100- μ s (c), and 5- μ s (d) stroboscopic illumination duration. (e) ACsN-denoised light-field image of (d). (f) The relationship between motion blur and SNR. The black dots represent the image SNR without ACsN, and the red dot marks the SNR of (e) with ACsN. (g, h) The 3D reconstruction of (d) and (e), respectively. (i) Focal images of (g) and (h), respectively. Scale bars: 10 μ m (a-e), 1 μ m (i).

4. From the raw LFC images (for example, Figs. 2a, 2e, 2f, 4g, 5a, 5b...), the signal-to-noise ratio appears to be enough for the subsequent deconvolution process. Therefore, it raises the question of whether the denoising algorithm is necessary at all times. It would be valuable to discuss the necessity of the denoising algorithm and examine whether there is any loss of resolution after denoising. I recommend that the authors could add an experimental comparison of LFC reconstruction results without and with pre-denoising, preferably at different noise levels.

Response: Indeed, denoising methods like low-pass filtering imply a tradeoff between feature preservation and noise canceling. Specifically, low-pass filtering requires a careful calibration of the optical transfer function (OTF) of the experimental setup in order to avoid the undesired loss of useful details, which results in image blurring. Even so, readout noise is white Gaussian noise, which is present at all frequencies and cannot be totally removed by low-pass filtering (but just blurred). On the other hand, the strength of the ACsN algorithm^{11,33} avoids this tradeoff by addressing the sparsity, which allows us to cancel the (low sparsity) readout noise generated by the camera while maintaining all the (high sparsity) features of the input signal. Thus, ACsN can filter out readout noise, preserving the fine details and resolution of signals comparable to the expected values of a noise-suppressed camera (with only intrinsic photon noise, whose magnitude is the square root of the intensity). While

SNR has always been the limitation to resolving fine details in digital imaging, with ACsN, such limitation has been pushed close to the ideal limit, applicable to the full SNR range of fluorescent detection.

Next, we would like to clarify that the images (**Figures 3a, 3e, 3f, 4g, 5a, and 5b**), as mentioned by the Reviewer, were the results already processed by ACsN. Their original raw images, however, have a significantly lower SNR. The denoising process is a necessary pre-processing step for the accuracy and quality of the accuracy of our subsequent 3D reconstruction. We have revised the captions (**Figures 3a, 3e, 3f, 4g, 5a, and 5b**) accordingly for clarity and added the corresponding unprocessed raw images to **Supplementary Figure 5** for readers' better information. Furthermore, as requested by the Reviewer, we have conducted additional imaging experiments. A comparison of the results with and without denoising verified the denoising algorithm enhanced image quality across a range of noise levels (**Figure R11**). We have added these results and quantitative analysis in the revised **Supplementary Figure 7** for readers' better information.

Figure R10. Comparisons of reconstructed image quality with and without denoising at different noise levels. (a, e, i) Light-field images of mitochondria in HeLa cells at different noise levels without denoising (left column) and their corresponding reconstructions (right column). (b, f, j) Zoomed-in regions marked with the dashed boxes in (a), (e), and (i), respectively, showing the intensity profiles along the dashed lines. (c, g, k) Light-field images of mitochondria in HeLa cells in (a, e, i) with denoising (left column) and their corresponding reconstructions (right column). (d, h, l) Zoomed-in region marked in the dashed boxes in (c), (g), and (k), respectively, showing better resolution of fine details. (m, o) Light-field images of peroxisomes (m) in HeLa cells and mitochondria (o) in Jurkat cells, respectively, at different noise levels without denoising (left column) and their reconstructions (right column). (n, p) Light-field images of peroxisomes (n) in HeLa cells and mitochondria (p) in Jurkat cells, respectively, at different noise levels with denoising (left column) and their reconstructions (right column). Scale bars: 10 μm (a, c, e, g, i, k, m, n, o, p), 1 μm (b, d, f, h, j, l).

5. The authors mentioned that the depth of field of LFC is 3-4 μm , which seems to be sufficient only for the observation of cultured adherent cells. However, for flowing cells, the depth-of-field range may not cover the entire cell due to the diameter of the microfluidic channel. Will this become a problem for practical application? How can it be solved? I think it is quite necessary for the 3D imaging of flowing cytometry. Or maybe people can directly use some methods with extended depth of field for 2D imaging without the requirement for axial information? The necessity of axial resolution has not been demonstrated.

Response: In this work, LFC presents a depth of field (DOF) of $\sim 6 \mu\text{m}$ for high-resolution light-field acquisition because of the hybrid PSF employed that addressed the spherical aberration so that extended layers in depth could be retrieved (**Figures 3, 4** and **Supplementary Note 7.2**). This DOF covers a significant thickness of cells and extends $>5\times$ compared to conventional epi-fluorescence acquisition (using the same $100\times$, 1.45NA objective lens). However, we agree with the Reviewer that such DOF may not be able to cover those cell types that may extend beyond $10 \mu\text{m}$ in thickness.

To enhance the coverage, three practical solutions (1-3) can be feasibly executed to extend the DOF in the further development of the LFC system. For the Reviewer's information, these solutions include the implementation of (1) a low-magnification objective lens, (2) an additional multi-focal MLA, and (3) an electrically tunable lens (illustrated in **Figure R11**). In particular, the initial alteration (1) involves transitioning to a $40\times$ objective lens (e.g., Nikon CFI Plan Fluor $40\times$, 1.3NA Oil). This switch is concomitant with adjustments in the design parameters for both the micro-lens array (MLA) and the Fourier lens³⁴ (**Figure R11a**). Here, we propose the parameters for the MLA ($f_{\text{ML}} = 55.8 \text{ mm}$, pitch $d = 3.3 \text{ mm}$, 7 hexagonal microlenses) and the Fourier lens ($f_{\text{FL}} = 150 \text{ mm}$). Based on our theoretical model described in **Supplementary Note 7.2**, these modifications lead to ~ 1.5 -fold improvement in the DOF (i.e., $\sim 8.2 \mu\text{m}$), $3\times$ expanded field of view ($220 \mu\text{m}$), 3D resolution of 600-850 nm and 1.1-1.5 μm in the lateral and axial dimensions, respectively. In the alternative solution (2), we propose a multi-focal Fourier light-field design to enhance the DOF by placing an additional Fourier light-field path with an offset of 40 mm away from the native image plane so that the two Fourier light-field paths simultaneously capture connective focal ranges (**Figure R11b**). Last but not least, the solution (3) will replace the normal tube lens with a focus tunable lens (e.g., an electrically tunable lens or ETL) so that different depth layers can be refocused corresponding to the focal changes of the ETL (**Figure R11c**). By synchronizing the focal scan of the ETL and camera frames, the DOF can be efficiently extended by accumulating the images acquired from multiple frames. In light of this comment, we have added the details and illustrations to the manuscript (**Supplementary Note 7.2**) for readers' information.

In fact, the third solution, using an ETL on a conventional imaging cytometer, can readily enhance the depth of field for 2D acquisition for applications without the need for axial information. However, technically, all biological samples are three-dimensional, especially when cells are placed in a fluidic condition and, as a result, appear to have a more native spheric morphology. As mentioned in **Ref 1**, the authors commented at the conclusion that *'Clearly, imaging flow cytometry has proven value in combining the advantages of a microscope and a flow cytometer. However, the technique does have limitations, for example, in lacking capability for: workflow automation, cell sorting, repeated time-lapse imaging of the same cell and 3D resolution ...'* We believe the advancement of IFC into higher spatiotemporal dimensions is of great demand and the dissemination of LFC will provide critical morphological details and a wide range of biological and translational implications.

Figure R11. Experimental diagrams (a-c) for the proposed solutions (1-3), respectively.

Overall, addressing these major points will strengthen the manuscript and help in evaluating its suitability for publication in Nature Communications.

Response: In response to the Reviewer's comments and recommendations, we believe the amendments and clarifications provided above meet the criteria for publication in Nature Communications. Additionally, we have made substantive advancements in our algorithmic framework, specifically incorporating deep neural networks for the task of image reconstruction. To elaborate, traditional Richardson-Lucy deconvolution algorithms have been replaced by deep-learning algorithms optimized for 3D light-field image retrieval. The comparative analysis reveals that our updated approach delivers results of high quality and fidelity, closely approximating the wide-field-based ground truth (**Figure R12**). These proof-of-concept findings have been briefly incorporated into the Discussion and Conclusion section as a future perspective of the approach, and the methodological details have been documented in revised **Supplementary Note 5**. Notably, the computational framework achieves over a 100-fold acceleration in the processing time, necessitating less than 0.06 seconds for rendering each cellular volume, an exceptional advance and quality for studying large cell populations with IFC. As a summary, we believe LFC features a novel and significant advance for highly desirable cytometric imaging techniques for a broad range of cell biological studies.

Figure R12. Deep learning-based 3D light-field image retrieval. For the training dataset, we utilized a collection of 500 previously acquired wide-field volumes featuring HeLa peroxisomes (a). These volumes were subjected to deconvolution (b) using a 3D wide-field PSF to enhance their SNR, thereby serving as our ground truth (GT). Subsequently, these deconvolved wide-field volumes were convolved with a 3D light-field PSF to generate synthetic light-field images. The resultant elemental images were segmented and compiled along the channel dimension to create the training input for the neural network (c). The network architecture employed is based on the U-Net framework, as depicted in (d). To accommodate the GPU memory constraints of our workstation, the training inputs were resized to dimensions of $512 \times 512 \times 3$ pixels, while the ground truths were resized to $512 \times 512 \times 64$ pixels. The voxel dimensions are set at $130 \text{ nm} \times 130 \text{ nm} \times 65 \text{ nm}$. The network underwent 500 training epochs, completed in an approximate time span of 5-6 hours, utilizing an Nvidia TITAN RTX graphics card for computation. The deep learning-generated reconstructions of (e, h, k) are presented in (f, i, l) and is compared with corresponding wide-field scanning results in (g, j) and Richardson-Lucy deconvolution (RLD) results in (m). The quality of the deep learning-reconstructed image is found to be comparable to that achieved through wide-field scanning results and deconvolved results. The intensity values were normalized to a 0-1 scale. The image quality of the deep learning results was measured with 3D structure similarity indices (3D SSIM) and peak signal-to-noise ratios (PSNR). Scale bars: $10 \mu\text{m}$ (c, e, h, k), $5 \mu\text{m}$ (f, g, i, j), $1 \mu\text{m}$ (l, m).

REFERENCES

- 1 Rees, P., Summers, H. D., Filby, A., Carpenter, A. E. & Doan, M. Imaging flow cytometry. *Nature Reviews Methods Primers* **2**, 86 (2022). <https://doi.org:10.1038/s43586-022-00167-x>
- 2 Almada, P. *et al.* Automating multimodal microscopy with NanoJ-Fluidics. *Nature Communications* **10**, 1223-1223 (2019). <https://doi.org:10.1038/s41467-019-09231-9>
- 3 AbuZineh, K., Joudeh, L. I., Al Alwan, B., Hamdan, S. M., Merzaban, J. S. & Habuchi, S. Microfluidics-based super-resolution microscopy enables nanoscopic characterization of blood stem cell rolling. *Science Advances* **4**, eaat5304-eaat5304 (2018). <https://doi.org:10.1126/sciadv.aat5304>
- 4 Mandracchia, B., Son, J. & Jia, S. Super-resolution optofluidic scanning microscopy. *Lab Chip* **21**, 489-493 (2021). <https://doi.org:10.1039/d0lc00889c>
- 5 Liu, W., Kim, G. R., Takayama, S. & Jia, S. Fourier light-field imaging of human organoids with a hybrid point-spread function. *Biosens Bioelectron* **208**, 114201 (2022). <https://doi.org:10.1016/j.bios.2022.114201>
- 6 Hua, X., Liu, W. & Jia, S. High-resolution Fourier light-field microscopy for volumetric multi-color live-cell imaging. *Optica* **8**, 614-620 (2021). <https://doi.org:10.1364/optica.419236>
- 7 Han, K. *et al.* 3D super-resolution live-cell imaging with radial symmetry and Fourier light-field microscopy. *Biomed Opt Express* **13**, 5574-5584 (2022). <https://doi.org:10.1364/BOE.471967>
- 8 Ravindran, R. *et al.* Peroxisome biogenesis initiated by protein phase separation. *Nature* **617**, 608-615 (2023). <https://doi.org:10.1038/s41586-023-06044-1>
- 9 Sugiura, A., Mattie, S., Prudent, J. & McBride, H. M. Newly born peroxisomes are a hybrid of mitochondrial and ER-derived pre-peroxisomes. *Nature* **542**, 251-254 (2017). <https://doi.org:10.1038/nature21375>
- 10 Barteneva, N. S. *et al.* *Imaging flow cytometry: methods and protocols.* (Springer, 2016).
- 11 Mandracchia, B., Hua, X., Guo, C., Son, J., Urner, T. & Jia, S. Fast and accurate sCMOS noise correction for fluorescence microscopy. *Nature Communications* **11**, 94-94 (2020). <https://doi.org:10.1038/s41467-019-13841-8>
- 12 Ling, Z., Han, K., Liu, W., Hua, X. & Jia, S. Volumetric live-cell autofluorescence imaging using Fourier light-field microscopy. *Biomedical Optics Express* **14**, 4237 (2023). <https://doi.org:10.1364/boe.495506>

- 13 Mikami, H. *et al.* Virtual-freezing fluorescence imaging flow cytometry. *Nat Commun* **11**, 1162 (2020). <https://doi.org:10.1038/s41467-020-14929-2>
- 14 Diebold, E. D., Buckley, B. W., Gossett, D. R. & Jalali, B. Digitally synthesized beat frequency multiplexing for sub-millisecond fluorescence microscopy. *Nature Photonics* **7**, 806-810 (2013). <https://doi.org:10.1038/nphoton.2013.245>
- 15 Holzner, G. *et al.* High-throughput multiparametric imaging flow cytometry: toward diffraction-limited sub-cellular detection and monitoring of sub-cellular processes. *Cell Rep* **34**, 108824 (2021). <https://doi.org:10.1016/j.celrep.2021.108824>
- 16 Fan, Y.-J. *et al.* Microfluidic channel integrated with a lattice lightsheet microscopic system for continuous cell imaging. *Lab on a Chip* **21**, 344-354 (2021).
- 17 Kumar, P., Joshi, P., Basumatary, J. & Mondal, P. P. Light sheet based volume flow cytometry (VFC) for rapid volume reconstruction and parameter estimation on the go. *Scientific reports* **12**, 1-15 (2022).
- 18 Schraivogel, D. *et al.* High-speed fluorescence image-enabled cell sorting. *Science* **375**, 315-320 (2022). <https://doi.org:10.1126/science.abj3013>
- 19 Quint, S. *et al.* 3D tomography of cells in micro-channels. *Applied Physics Letters* **111**, 103701 (2017).
- 20 Merola, F. *et al.* Tomographic flow cytometry by digital holography. *Light Sci Appl* **6**, e16241 (2017). <https://doi.org:10.1038/lsa.2016.241>
- 21 Kleiber, A., Ramoji, A., Mayer, G., Neugebauer, U., Popp, J. & Henkel, T. 3-Step flow focusing enables multidirectional imaging of bioparticles for imaging flow cytometry. *Lab on a chip* **20**, 1676-1686 (2020).
- 22 Han, Y. *et al.* Cameraless high-throughput three-dimensional imaging flow cytometry. *Optica* **6**, 1297-1304 (2019).
- 23 Zhang, Z. *et al.* A high-throughput technique to map cell images to cell positions using a 3D imaging flow cytometer. *Proc Natl Acad Sci U S A* **119**, e2118068119 (2022). <https://doi.org:10.1073/pnas.2118068119>
- 24 Miura, T., Mikami, H., Isozaki, A., Ito, T., Ozeki, Y. & Goda, K. On-chip light-sheet fluorescence imaging flow cytometry at a high flow speed of 1 m/s. *Biomedical optics express* **9**, 3424-3433 (2018).
- 25 Nitta, N. *et al.* Intelligent Image-Activated Cell Sorting. *Cell* **175**, 266-276 e213 (2018). <https://doi.org:10.1016/j.cell.2018.08.028>
- 26 Holzner, G. *et al.* High-throughput multiparametric imaging flow cytometry: toward diffraction-limited sub-cellular detection and monitoring of sub-cellular processes. *Cell Reports* (2021). <https://doi.org:10.1016/j.celrep.2021.108824>
- 27 Isozaki, A. *et al.* Intelligent image-activated cell sorting 2.0. *Lab on a Chip* **20**, 2263-2273 (2020). <https://doi.org:10.1039/d0lc00080a>
- 28 Munoz, H. E. *et al.* Single-Cell Analysis of Morphological and Metabolic Heterogeneity in *Euglena gracilis* by Fluorescence-Imaging Flow Cytometry. *Anal Chem* **90**, 11280-11289 (2018). <https://doi.org:10.1021/acs.analchem.8b01794>
- 29 Rane, A. S., Rutkauskaite, J., deMello, A. & Stavrakis, S. High-throughput multi-parametric imaging flow cytometry. *Chem* **3**, 588-602 (2017).
- 30 Ota, S. *et al.* Ghost cytometry. *Science* **360**, 1246-1251 (2018). <https://doi.org:10.1126/science.aan0096>

- 31 George, T. C. *et al.* Distinguishing modes of cell death using the ImageStream® multispectral imaging flow cytometer. *Cytometry Part A: the journal of the International Society for Analytical Cytology* **59**, 237-245 (2004).
- 32 Goda, K. *et al.* High-throughput single-microparticle imaging flow analyzer. *Proc Natl Acad Sci U S A* **109**, 11630-11635 (2012). <https://doi.org:10.1073/pnas.1204718109>
- 33 Strack, R. Say goodbye to sCMOS noise. *Nature Methods* **17**, 252-252 (2020). <https://doi.org:10.1038/s41592-020-0790-3>
- 34 Guo, C., Liu, W., Hua, X., Li, H. & Jia, S. Fourier light-field microscopy. *Optics Express* **27**, 25573-25594 (2019). <https://doi.org:10.1364/OE.27.025573>

REVIEWER COMMENTS

Reviewer #1 (Remarks to the Author):

Thank the authors for their response. I have noted that the authors have included additional experiments to address the concerns I raised. However, my primary concerns still remain with regards to its resolution for subcellular structure imaging and the lack of applications to support the claimed advantages and significance of high cell throughput. In their response, the authors clarify that compared to previous IFC systems, the LFC system offers improved resolution and cell throughput. While this is a positive development, there is no data to demonstrate how exactly these improvements translate into practical benefits for researchers using this technology.

(1) The authors claim that the spatial resolution of the LFC system is 300-600nm, but the resolution presented in the manuscript is 400-600nm (Figure 3). Why?

(2) The key to study fine subcellular structures in single cells is to achieve high resolution (50-100nm resolution for super-resolution systems and about 250nm resolution for confocal microscopy). The resolution of the LFC systems (400-600nm) is clearly not enough to distinguish the subcellular structures. The authors claim that it can provide unprecedented high cell throughput. However, the significance of high cell throughput for the study of subcellular structures in single cells is still not shown in the manuscript and the response letter.

(3) Mitochondria and peroxisomes in Hela cells were imaged using the LFC system. Please explain why the image resolution in Figure R1 is significantly higher than that in Figure 3.

(4) Many experimental data added in the response letter are not shown in the manuscript and SI, and it is recommended to include this part of the data in the manuscript.

Reviewer #2 (Remarks to the Author):

I am satisfied that the authors have modified the manuscript to account for my concerns and the revised document is significantly improved given their modifications in response to the other reviewers. I am happy to recommend publication in Nature Communications.

Reviewer #3 (Remarks to the Author):

As already mentioned in my previous review, this work presents a new method for 3D imaging flow cytometry using light-field microscopy (LFM). The proposed technique is shown to be effective in a variety of conditions. The article is well-written and well-organized with relevant references.

My initial concerns about the first draft are the technical advantages of this method compared with their previous work and several technique concerns. The authors have conducted substantial new experiments to address my concerns and further show their main contributions in optimizing their Fourier LFM for IFC applications with critical improvement in the cell throughput, which is quite impressive. Overall, I recommend this work for publication.

I have another point of interest. As other reviewers have mentioned, resolution is one of the concerns for organelle imaging. I was wondering if it is possible to apply light-field super-resolution algorithms (e.g., VsLFM or HyLFM) or digital adaptive optics to enhance the resolution during the post-processing process. I think at least the authors can briefly discuss this potential in the discussion.

RESPONSE TO REVIEWERS' COMMENTS

We thank the Reviewers for thoroughly examining the manuscript and providing positive feedback. Here, we submit a point-by-point response letter in which we have provided detailed responses to each comment from the Reviewers and made corresponding revisions in our revised manuscript as presented in the following.

Response to Reviewer #1

Thank the authors for their response. I have noted that the authors have included additional experiments to address the concerns I raised. However, my primary concerns still remain with regards to its resolution for subcellular structure imaging and the lack of applications to support the claimed advantages and significance of high cell throughput. In their response, the authors clarify that compared to previous IFC systems, the LFC system offers improved resolution and cell throughput. While this is a positive development, there is no data to demonstrate how exactly these improvements translate into practical benefits for researchers using this technology.

Response: We thank the Reviewer for the feedback and are pleased that our additional experimental results have addressed the concerns the Reviewer previously raised.

Here, we would like to clarify the remaining confusion. First, we would like to elaborate on the physical novelty by which LFC overcomes the trade-off between resolution and throughput inherent in existing IFC systems. High-resolution imaging provides detailed insights into subcellular structures, yet this is often at the cost of throughput. Conversely, systems tailored for enhanced throughput may compromise their resolution. Theoretically, analytical throughput decreases in quadratic proportion to the increase in magnification (i.e., resolution) due to constraints imposed by the effective pixel size and the maximal flow velocity that precludes motion blur (see **Figure 2C** in Ref. 1). Conventional approaches have typically achieved high- or super-resolution imaging by considerably restricting throughput²⁻⁴. However, LFC proposes a significant advance by combining the 100× objective lens with individual microlenses, thus formulating an effective magnification of 42.5× (thereby enhancing the throughput over a conventional 100× system). Then, LFC restores the near-diffraction-limited resolution, characterized by the 100× objective lens, through the wave-optics-based reconstruction of elemental light-field images. This combinatorial strategy represents a substantial advance in alleviating the resolution-throughput tradeoff for IFC while retaining the unique snapshot 3D ability of light-field imaging, which, as a result, collectively surpasses the analytical throughput of conventional wide-field systems (**Figure RR1**). We have also added a detailed discussion and illustration to the revised **Main Text** (the **Discussion** section) and **Supplementary Note 7.6**.

Figure RR1. Variation in the analytical throughput of LFC and conventional 2D wide-field cytometry (WFC) as a function of objective magnification (10×, 15×, 20×, 40×, 100×). In addition to the 3D ability, LFC achieves a higher throughput owing to the microlens array, which results in an effectively lower magnification. The reconstruction of elemental images allows for the recovery of the full resolution.

Figure redacted.

Figure RR2.

Next, we would like to expand upon the applicability of LFC. It is noteworthy that the light-field cytometer is implemented seamlessly with both **standard epi-fluorescence microscopy** and **microfluidics**. This quality facilitates **a platform of ready and high accessibility and relevance** to immediately transform broad existing IFC studies. For example, in this work, we have demonstrated various applications such as particle sorting, cell morphology screening, cell heterogeneity, chemical-induced cell apoptosis, and lipid nanoparticle-enclosed mRNA delivery. The resolution (400-600 nm) has been validated by visualizing subcellular structures such as nuclei, mitochondria, peroxisomes, cytoplasm, and membranes. These demonstrations will contrast existing 3D IFC applications of similar subcellular entities, typically relying on a spatial resolution of 800-1800 nm⁵⁻⁷. Furthermore, LFC will enable the visualization of many other subcellular features that require IFC, such as spot counting for nanoparticle uptake in cells, calcium location detection in T cells, the activation of eosinophils, blood cell classification, and micronucleus phenotypes identification in cells exposed to a genotoxic compound⁸. For better clarity, we have added further elaborations in the revised **Supplementary Note 11**.

In particular, to specify an emerging example of the applicability, LFC has recently been utilized to examine **the delivery of mRNA** using lipid nanoparticles (LNPs). The work, entitled *Cationic cholesterol-dependent LNP delivery to the liver, heart, and lung*, was led by our co-author and collaborator, Professor James Dahlman (Emory University). We have included a pre-print figure on the above page (**Figure RR2**) for the editorial review. LFC provides the first-of-its-kind 3D images of LNP⁺-targeted liver, heart, and lung cells, and the results evidence that charge-dependent tropism holds promise for genetic diseases that require tissue-specific delivery. The statistical results are consistent with traditional fluorescence-activated cell sorting (FACS)⁹. We expect this state-of-the-art application to clarify the Reviewer's concern about the potential usability of LFC for single-cell analysis. Based on these first impactful demonstrations of LFC, the system is anticipated to be highly adaptable and thus transform a broader range of research upon acceptance and publication in *Nature Communications*.

Major concerns:

(1) The authors claim that the spatial resolution of the LFC system is 300-600nm, but the resolution presented in the manuscript is 400-600nm (Figure 3). Why?

Response: We apologize for the confusion and would like to clarify these reported measurements. With the phantom samples (e.g., microspheres), the higher signal-to-noise ratio enables us to achieve a slightly better spatial measurement between 300-600 nm (we only mentioned this quantification in Figure 2 of phantom results). In the context of biological specimens, we have validated the consistent spatial resolution of 400-600 nm. To avoid confusion, we have explicitly emphasized in **the caption of Figure 2** and decided to report the resolution of 400-600 nm in accordance with the biological validation throughout the Abstract, Main Text, and Supplementary Information for readers' clarity and more accurate information.

(2) The key to study fine subcellular structures in single cells is to achieve high resolution (50-100nm resolution for super-resolution systems and about 250nm resolution for confocal microscopy). The resolution of the LFC systems (400-600nm) is clearly not enough to distinguish the subcellular structures. The authors claim that it can provide unprecedented high cell throughput. However, the significance of high cell throughput for the study of subcellular structures in single cells is still not shown in the manuscript and the response letter.

Response: Indeed, LFC offers a relatively lower resolution compared with super-resolution and confocal systems. On the other hand, as addressed in our previous response letter, we emphasized that such resolving power remains markedly capable of revealing a substantial range of subcellular features, such as size, shape, biomarker intensity, physiological state, and other morphological and biochemical characteristics¹⁰.

Next, we would like to address the **significance of high cell throughput** for single-cell studies. In essence, **high throughput is one primary advantage of IFC over traditional single-cell imaging platforms**. This major

advantage leads to processing and analyzing thousands to millions of cells in a single experiment, orders of magnitude higher than conventional imaging techniques. Specifically, high throughput offers high content, multiparametric analysis, and statistical significance for large-scale cell studies and screening applications. These single-cell details allow for identifying genes, pathways, and cell biological mechanisms at *the population level* underlying disease diagnosis in clinical settings^{11, 12}. The high number of cells analyzed per sample increases the statistical power and reduces the impact of sample bias in the experiments, which is crucial for detecting subtle and rare phenotypic changes and for robust data interpretation in biological research. The automated nature of cytometric imaging allows for rapid sample loading, data acquisition, and analysis, reducing the time and labor required for experiments. This efficiency is vital in high-throughput screening and large-scale studies. The throughput of IFC also enables cell studies that necessitate imaging of fresh clinical samples or in their native state post-extraction from organs (e.g., **Figure 6**). The high throughput system provides better integration with other technologies, such as mass spectrometry or genomics platforms, providing a more comprehensive analysis of the cellular state. Relevant to all the above cases, confocal or super-resolution microscopy becomes limited. As mentioned in Rees, et al, Nature Review Methods Primers 2022⁸, “Future iterations may bring novel data acquisition and sorting technologies at **higher resolution**, with **higher dimensions** (larger 2D/3D FOV, temporal feature availability), while retaining, if not improving, the **high throughput** that makes imaging flow cytometry advantageous over other single-cell imaging platforms.”

In light of this comment regarding the significance of high cell throughput for single-cell studies, we have added explicit and considerable statements and references in the revised **Supplementary Note 7.6**.

(3) Mitochondria and peroxisomes in HeLa cells were imaged using the LFC system. Please explain why the image resolution in Figure R1 is significantly higher than that in Figure 3.

Response: We apologize for the confusion and would like to clarify the difference between **Figure R1** (already added to **Supplementary Note 8**) and **Figure 3**.

(i) The review experiments (**Figure R1**) were performed in a culture dish, purposefully designed to allow for cell-to-cell identification and comparison with other modalities, including wide field and structured illumination microscopy. Therefore, the cells were attached to the substrate and exhibited flat-distributed subcellular morphology in the condition of **Figure R1**. In contrast, HeLa cells in **Figure 3** were captured in the microfluidic environment, where the flowing cells display their native spherical morphology. Mitochondria and peroxisomes are volumetric and axially stacked in this condition, complicating the resolved subcellular structures.

(ii) The exposure times were different in the two cases. In **Figure 3**, the cells were imaged at high throughput with a short exposure time of 100 μ s. In **Figure R1**, the images were acquired with a 1000 \times longer exposure time of 100 ms to match the imaging condition of other comparative modalities. Different exposure times resulted in different signal levels collected and signal-to-noise ratios (SNRs), thus leading to variations in image visualization.

(iii) The cells were prepared using different staining methods, i.e., GFP- or tracker-labeled and immuno-staining, for **Figure 3** and **Figure R1**, respectively (see **Supplementary Note 13**), which resulted in relatively higher SNR in **Figure R1**.

(iv) The spherical aberrations of flowing cells moderately degrade the overall resolution and image quality in **Figure 3**, considering the refractive index mismatch for the high-NA objective lens. In response to this point, we performed numerical simulations to show the influence of aberration on the LFC results (**Figure RR3**). As seen, the simulated Siemens stars at different axial positions (**Figure RR3a**) were convolved with PSFs taken at the surface of the substrate (approximating the condition in **Figure R1**) and at 10 μ m deep into the solution (approximating the condition in **Figure 3**), respectively, to generate synthesized light-field images (left columns in **Figure RR3b** and **RR3c**). The synthesized light-field images were then deconvolved with the corresponding

PSFs to get the reconstructed volumes. The image quality was quantitatively measured with the reconstructed depth layers at the corresponding axial positions (middle and right columns in **Figure RR3b** and **RR3c**). At each axial position, the intensity profiles along the dashed circles of the same size in both conditions were plotted for comparison. As shown in **Figure RR3d**, the intensity profiles of the sample at the surface show higher contrast and slightly resolution compared to the sample 10 μm deep in the solution. We have now added explicit statements in the revised **Supplementary Note 8**.

Figure RR3. Simulation of Siemens stars at different axial locations in the solution. (a) Illustrations of Siemens star patterns at various axial positions away from the focal plane. (b, c) Raw light-field images (left columns), the corresponding z layers of 3D reconstructions (middle columns) with zoomed-in region within the solid-line circles (right columns) of the Siemens stars at the surface of the substrate (b) and 10 μm deep into the solution (c). (d) Intensity profiles along the dashed circles in (b) and (c). Scale bars: 10 μm (b, c, left and middle columns), 1 μm (b, c, right columns).

(4) Many experimental data added in the response letter are not shown in the manuscript and SI, and it is recommended to include this part of the data in the manuscript.

Response: We have confirmed that, in the previously revised manuscript, all the additional data shown in the previous response letter, except for **Figure R7**, have already been included. The corresponding locations of the revisions in the manuscript are listed in **Table RR1** below. We have now added **Figure R7** to **Figure S12**. Furthermore, the data provided in this current response letter have also been inserted into the corresponding sections, as indicated in **Table RR1**.

Table RR1 (new revisions made in this current response letter marked in red)	
Response letters	Revised manuscript
Figure R1	Supplementary Note 8
Figure R2	Supplementary Note 10
Figures R3 and R9	Figure S19 in Supplementary Note 6

Figure R4	Supplementary Note 11
Figure R5	Figure S16 in Supplementary Note 5
Figure R6	Figure S4
Figure R7	Figure S12
Table R1	Table S2 in Supplementary Note 11
Figure R8 and R9	Figure S15 in Supplementary Note 5
Figure R10	Figure S7
Figure R11	Figure S20 in Supplementary Note 7.2
Figure R12	Figure S17 in Supplementary Note 5
Figure RR1	Figure S23 in Supplementary Note 7.6
Figure RR2	Adapted from Radmand, et al, in press
Figure RR3	Figure S25 in Supplementary Note 8
Figure RR4	Figure S21

Response to Reviewer #2

I am satisfied that the authors have modified the manuscript to account for my concerns and the revised document is significantly improved given their modifications in response to the other reviewers. I am happy to recommend publication in Nature Communications.

Response: We appreciate the Reviewer's recommendation for publication in *Nature Communications*.

Response to Reviewer #3

As already mentioned in my previous review, this work presents a new method for 3D imaging flow cytometry using light-field microscopy (LFM). The proposed technique is shown to be effective in a variety of conditions. The article is well-written and well-organized with relevant references.

My initial concerns about the first draft are the technical advantages of this method compared with their previous work and several technique concerns. The authors have conducted substantial new experiments to address my concerns and further show their main contributions in optimizing their Fourier LFM for IFC applications with critical improvement in the cell throughput, which is quite impressive. Overall, I recommend this work for publication.

I have another point of interest. As other reviewers have mentioned, resolution is one of the concerns for organelle imaging. I was wondering if it is possible to apply light-field super-resolution algorithms (e.g., VsLFM or HyLFM) or digital adaptive optics to enhance the resolution during the post-processing process. I think at least the authors can briefly discuss this potential in the discussion.

Response: We appreciate the Reviewer's recommendation for publication in *Nature Communications*.

Indeed, as the Reviewer pointed out, we agree with the feasibility of applying advanced strategies, such as light-field super-resolution algorithms (e.g., VsLFM¹³ or HyLFM¹⁴) or digital adaptive optics (Wu, et al. Cell, 2021¹⁵) to enhance the resolution during the post-processing process. Specifically, VsLFM enhances resolution by

leveraging multiple angles of view scanned by piezo-steering mirrors combined with a digital adaptive optics algorithm. In addition, HyLFM employs light-sheet illumination, facilitating the simultaneous capture of high-resolution images that serve as the training and validation datasets for deep learning networks. We expect these techniques to enhance the resolution of LFC. We have added these discussions to **Supplementary Note 7.1**. Lastly, we illustrated these potential strategies as integrated into the LFC system for future development (**Figure RR4**). It should be noted that these methods would be the first integration into Fourier light-field and cytometric imaging settings. We have expanded relevant references in the **Discussion** section in the **Main Text** and added revised **Figure S21** based on **Figure RR4**.

Figure RR4. (a) Scheme of integrating VsLFM to LFC, implementing piezo-steering mirrors into the Fourier light-field acquisition. The tomographic images taken will be sent to a deep learning network for training so the scanning images can eventually be virtually predicted. (b) Scheme of integrating HyLFM to LFC, which concomitantly acquires light-sheet image stacks serving as continuous training and validation for deep learning network reconstructing the LFC data.

REFERENCES

1. Holzner, G. et al. High-throughput multiparametric imaging flow cytometry: toward diffraction-limited sub-cellular detection and monitoring of sub-cellular processes. *Cell Rep* **34**, 108824 (2021).
2. AbuZineh, K. et al. Microfluidics-based super-resolution microscopy enables nanoscopic characterization of blood stem cell rolling. *Sci Adv* **4**, eaat5304 (2018).
3. Almada, P. et al. Automating multimodal microscopy with NanoJ-Fluidics. *Nat Commun* **10**, 1223 (2019).
4. Mandracchia, B., Son, J. & Jia, S. Super-resolution optofluidic scanning microscopy. *Lab Chip* **21**, 489-493 (2021).
5. Kumar, P., Joshi, P., Basumatary, J. & Mondal, P.P. Light sheet based volume flow cytometry (VFC) for rapid volume reconstruction and parameter estimation on the go. *Scientific reports* **12**, 1-15 (2022).
6. Fan, Y.-J. et al. Microfluidic channel integrated with a lattice lightsheet microscopic system for continuous cell imaging. *Lab on a Chip* **21**, 344-354 (2021).
7. Miura, T. et al. On-chip light-sheet fluorescence imaging flow cytometry at a high flow speed of 1 m/s. *Biomedical optics express* **9**, 3424-3433 (2018).
8. Rees, P., Summers, H.D., Filby, A., Carpenter, A.E. & Doan, M. Imaging flow cytometry. *Nature Reviews Methods Primers* **2**, 86 (2022).
9. Paunovska, K. et al. Nanoparticles Containing Oxidized Cholesterol Deliver mRNA to the Liver Microenvironment at Clinically Relevant Doses. *Adv Mater* **31**, e1807748 (2019).
10. Barteneva, N.S. et al. Imaging flow cytometry: methods and protocols. (Springer, 2016).
11. Gérard, A. et al. High-throughput single-cell activity-based screening and sequencing of antibodies using droplet microfluidics. *Nature Biotechnology* **38**, 715-721 (2020).
12. Boutros, M., Heigwer, F. & Laufer, C. Microscopy-Based High-Content Screening. *Cell* **163**, 1314-1325 (2015).
13. Lu, Z. et al. Virtual-scanning light-field microscopy for robust snapshot high-resolution volumetric imaging. *Nature Methods* **20**, 735-746 (2023).
14. Wagner, N. et al. Deep learning-enhanced light-field imaging with continuous validation. *Nature Methods* **18**, 557-563 (2021).
15. Wu, J. et al. Iterative tomography with digital adaptive optics permits hour-long intravital observation of 3D subcellular dynamics at millisecond scale. *Cell* **184**, 3318-3332 e3317 (2021).

REVIEWERS' COMMENTS

Reviewer #1 (Remarks to the Author):

In the revised manuscript, the authors have added new data and properly addressed the issues I arose. I recommend this work for publication.

Reviewer #3 (Remarks to the Author):

The authors have fully addressed my concerns. I highly recommend their publication in Nature Communications.